# A Market for Accuracy: Classification under Competition

Ohad Einav [1]   Nir Rosenfeld [1]

## Abstract

Machine learning models play a key role for service providers looking to gain market share in consumer markets. However, traditional learning approaches do not take into account the existence of additional providers, who compete with each other for consumers. Our work aims to study learning in this market setting, as it affects providers, consumers, and the market itself. We begin by analyzing such markets through the lens of the learning objective, and show that accuracy cannot be the only consideration. We then propose a method for classification under competition, so that a learner can maximize market share in the presence of competitors. We show that our approach benefits the providers as well as the consumers, and find that the timing of market entry and model updates can be crucial. We display the effectiveness of our approach across a range of domains, from simple distributions to noisy datasets, and show that the market as a whole remains stable by converging quickly to an equilibrium.

## 1. Introduction

Machine learning models play an essential role in consumer markets of today. Across many domains, firms that offer products and services can significantly gain by attaining better predictions about their users, or providing better predictions for them. In a sense, this has made accuracy itself a commodity which consumers pursue; consider how user choices have come to depend on the quality of personalized predictions in recommendation systems, media platforms, online marketplaces, or health analytics services. The increasing demand for accurate predictions incentivizes firms to improve their predictions, which in turn creates a 'supply' of accuracy—a process which results in the formation of what we refer to as *accuracy markets*.

In accuracy markets, firms seek to maximize their market share by competing with other firms over who provides more users with accurate predictions. This paper aims to study such markets from three perspectives, namely of: (i) the firms (or '*learners*') that compete in the market, (ii) the population of potential users, and (iii) the market itself. We follow the market model proposed by Ben-Porat & Tennenholtz (2017; 2019), but focus on classification (rather than regression), which we argue is more natural for this setting. The main idea is that users choose a firm if it provides them with accurate predictions; if multiple such firms exist, then ties are broken randomly. When there is only one firm in the market (a monopoly), then maximal market share can be attained by maximizing accuracy directly—which is also optimal for users. However, once there is competition, each firm must take into account the predictions of all other firms, as it depends on their choice of classifiers. This dependence forms an oligopoly market in which the interests of the firms no longer necessarily align with user welfare, defined as the ratio of users for whom at least one firm is accurate.

The main result of Ben-Porat & Tennenholtz (2019) is that if firms in such markets can successfully compute a best (or better) response (i.e., find a model that maximizes market share conditioned on all other models remaining fixed), then dynamics will converge to a pure Nash equilibrium. This is insightful, but leaves many questions unanswered: When can best-response classifiers be found efficiently, and how?[1] What kind of equilibria can be reached, and what will happen on the way there? How will the market be shared by the different firms? And will outcomes be beneficial for users? Our goal is to shed light on these issues and others, using theoretical analysis and empirical evaluation, and as they relate to the learning firms, the users, and the market.

From the perspective of the learner, our results show that finding the best-response classifier is as hard—but also not harder than—solving a standard classification problem. In fact, given the predictions (or classifiers) of all other players, maximizing market share corresponds to maximizing a particular weighted accuracy objective. This means that although solving this exactly is hard, standard learning techniques (e.g., using proxy losses) can be highly effective in

---

[1] Faculty of Computer Science, Technion - Israel Institute of Technology. Correspondence to: Nir Rosenfeld <nirr@cs.technion.ac.il>.

*Proceedings of the 42nd International Conference on Machine Learning*, Vancouver, Canada. PMLR 267, 2025. Copyright 2025 by the author(s).

---

[1] Ben-Porat & Tennenholtz (2019) give an algorithm that applies to linear regression and relies on integer linear programming.

practice. It also provides the learner flexibility in choosing the model class to work with, as it sees fit under standard considerations (e.g., data size, compute capacity). Interestingly, however, we see that the choice of *when* to respond bears significant implications on outcomes for the learner. This brings to focus questions regarding the pioneering of a new market, entering an existing one, and maintaining user loyalty (or its harmful counterpart of user lock-in practices).

From the perspective of the market, our analysis suggests that most markets will exhibit strong anti-coordination (the other alternative being the existence of a single dominant strategy, which is possible but rare). In particular, market forces push firms to secure exclusive access to certain user sectors, and firms compete over who secures the larger sectors. Notably, gaining access *first* does not guarantee exclusivity; in fact, for simple markets with two firms we show that firms engage in a chicken-like game, where moving first is disadvantageous. Our analysis reveals that, despite competition, firms are in a sense *cooperating*: when one firms acts to increase its market share, this also serves to increase the market shares of the other. Empirically, we observe that for larger markets with more firms outcomes are more nuanced, although the order of play remains highly significant.

From the perspective of users, a direct result of the market is that competition improves welfare. What may be surprising is how efficient the market is: Empirically, we observe that welfare increases quickly and attains the maximum possible value with only a few firms, and after one round of updates. For the latter, we give theoretical grounding for why this can happen. One reason is that our model for the market enables efficient outcomes to materialize—as long as information flows freely. This has policy implications: a social planner that seeks to maximize welfare should incentive firms (or introduce regulation) to make their models public. Thus, transparency becomes an operational consideration which, in a utilitarian sense, works in favor of both firms *and* users.

We end with a series of experiments using synthetic and real data that demonstrate the underlying mechanics of accuracy markets and how they operate. Our results demonstrate that learning in such markets can be feasible, that competition converges quickly, and that the market is typically highly efficient and favorable to users. Results also highlight the importance of adjusting the objective to account for competition, and show how lacking to do so (and optimizing accuracy naïvely) can be detrimental to both firms and users. For the market, we present analysis revealing how it decomposes across firms, and measure concentration and market power. We also show the importance of timing market entry and model updates, the relation between performance and model class capacity, and the constructive role of information sharing. These results underscore the dynamics of accuracy markets and showcase the importance of adapting to competition.

## 2. Related work

Studying the dynamics of machine learning models competing for market share has been a budding line of research. Ben-Porat & Tennenholtz (2017; 2019) present a regression learning task where providers wish to maximize their market share of users by reducing prediction errors to below a given threshold. Their focus is on the equilibrium dynamics in the induced game between the providers. Employing a similar setting, Jagadeesan et al. (2024) focus on the effects of competition dynamics on social welfare, showing that better data representation does not necessarily translate to better welfare; some of our results echo theirs. Feng et al. (2022) study the bias-variance tradeoff in competitive settings. Yao et al. (2023; 2024a;b) introduce a competition setting for content creation and study welfare, equilibrium behaviors, and best-response dynamics. Ginart et al. (2021), Dean et al. (2024), and Su & Dean (2024) study how competitors specialize when user choices influence the observable data of each competitor. Our work puts emphasis on the learning task itself, and studies its effects on market dynamics and outcomes.

More generally, our research relates to the growing literature on strategic learning. The majority of work in this field models users as strategic agents that can manipulate their features (e.g., Hardt et al., 2016; Levanon & Rosenfeld, 2021). In contrast, we model users as choosing among alternatives, and put emphasis on the strategic role that learning must assume to contend with competition. The idea that users can choose a provider has been considered in Koren (2023) and Horowitz et al. (2024), but only for binary choice (i.e., join or drop out) and under uncertainty. Our work differs in that it supports choices between multiple firms in a competitive market. In a recent paper, Chen et al. (2025) study strategic learning with externalities; interestingly, their construction also gives rise to a potential game (as ours does), but between users (rather than providers). Our work also draws connections to the field of performative prediction (Perdomo et al., 2020), which studies how (re)training models can gradually change the underlying data distribution. As we show, this perspective applies to our approach when considered from the viewpoint of a single competing provider.

## 3. Setup

In our competitive learning setting, users are described by features $x \in \mathcal{X}$ and labels $y \in \mathcal{Y}$, over which there is an unknown joint distribution $p(x, y)$. There are $n$ service providing firms, $s_1, \ldots, s_n$, who provide prediction services to users, and together form a market. Given a training set $S = \{(x_i, y_i)\}_{i=1}^{m}$, each service provider $s_i$ learns a classi-

fier $h_i$ from some model class $H_i$. Each user $x$ then chooses a provider among those offering an accurate prediction:

$$s(x) \in \{s_i \; : \; h_i(x) = y\} \tag{1}$$

If multiple providers are accurate, then $s(x)$ is determined by a random tie-breaking rule. When no providers offer an accurate prediction, we denote the null choice by $s(x) = \varnothing$. Welfare is defined as the ratio of users that obtain service:

$$W(\boldsymbol{h}) = \mathbb{E}_p[\mathbb{1}\{s(x) \geq 1\}] \tag{2}$$

where $\boldsymbol{h} = (h_1, \ldots, h_n)$ are all classifiers in the market.

The goal of service providers is to maximize their market share, defined as the expected ratio of users that choose them. For each provider $s_i$, this depends on its choice of learned model $h_i$, but also on the set of all other models, $h_{-i}$. Formally, the market share of service provider $s_i$ is defined as:

$$\mu_i = \mu(h_i \mid h_{-i}) = \mathbb{E}_p[\mathbb{P}[s(x) = s_i]] \tag{3}$$

where probability is w.r.t. how $s(x)$ is chosen from the set of accurate providers in Eq. (1). For simplicity we assume ties are broken uniformly at random, namely $\mathbb{P}[s(x) = s_i] = 1/\kappa(x)$ where $\kappa(x) = |\{s_j : s(x) \in s_j\}|$. This conforms to the setting of Ben-Porat & Tennenholtz (2017; 2019).

When the market includes only a single provider, maximizing market share is equivalent to maximizing accuracy:

$$\mathrm{argmax}_{h \in H} \; \mathbb{E}_p[\mathbb{1}\{y = h(x)\}] \tag{4}$$

which is the standard objective of supervised learning, and in this case also maximizes welfare by definition. However, once there is competition, this connection breaks since each provider's market share becomes dependent on all others. Thus, the naïve approach of maximizing accuracy as a proxy becomes suboptimal, and the question of how providers maximize their market share must be considered jointly.

**Competitive learning as a game.** For a given distribution $p$, if we think of each provider $s_i$ as a player and interpret Eq. (3) as their utility, then this defines a game, which we refer to as an *accuracy game*. The strategy space for each $s_i$ is the set of all models in its model class $H_i$, and each tuple $\boldsymbol{h} = (h_1, \ldots, h_n)$ defines a game state. We will assume the game is played on the empirical distribution induced by $S$, but that final payoffs are given by expected market share w.r.t. $p$. Note the game remains well-defined when players have their own $S_i \sim p_i$, although current equilibrium results do not hold for this setting. In terms of information, we work in the full information setting where all players have complete access to the payoff matrix. However, as we will see, optimal strategies for providers require strictly less information.

**Dynamics.** To understand how game states progress, we will explore dynamics in which providers can update their predictive models over time, and in response to others. Assume w.l.o.g. that providers are ordered, $s_1 \prec s_2 \prec \cdots \prec s_n$. Then at round $t$, each provider in turn chooses their $h_i$ by playing *best response*, defined as:

$$h_i^t = \mathrm{BR}(h_{-i}^t) = \underset{h \in H_i}{\mathrm{argmax}} \; \mu(h \mid h_{-i}^t) \tag{5}$$

That is, providers respond by choosing the optimal classifier $h_i$ assuming all others classifiers remain fixed, namely $h_{-i}^t = (h_1^t, \ldots h_{i-1}^t, h_{i+1}^{t-1}, h_n^{t-1})$ and for some choice of initial classifiers $\{h_i^0\}$. We refer to $h_i^t$ as the *best-response classifier* of $s_i$, but note it need not be unique. Since solving Eq. (5) can be computationally infeasible, we will also consider approximate best responses that replace $\mu(h \mid h_{-i}^t)$ with a tractable surrogate objective (see Sec. 5).

**Equilibrium.** We will be interested in studying the game's equilibria, focusing mostly on pure Nash equilibrium (PNE). These are defined as states $\boldsymbol{h}$ in which no provider has incentive to unilaterally deviate from its chosen strategy:

$$\forall \, i \in [n], \; h' \in H_i : \quad \mu(h_i \mid h_{-i}) \geq \mu(h' \mid h_{-i}) \tag{6}$$

Ben-Porat & Tennenholtz (2017) prove that the game is a type of *potential game* (Monderer & Shapley, 1996).[2] Since the game is played on the empirical distribution, which implies that the set of all possible predictions is finite (even if $H$ is not), a direct result is that a PNE exists and is reachable via a finite sequence of best responses (Eq. (5)). Note that multiple equilibria may exist, and that these may differ significantly in market shares, market concentration, and induced welfare. Furthermore, not all equilibria can necessarily be reached via best-response dynamics, and the equilibrium that is reached can depend on the initial game state (i.e., choice of first classifiers) and the order of play.

## 4. Analysis

In this section we set out to analyze basic properties of accuracy markets. To permit tractable analysis, here we focus on simple two player markets. We start with a restricted model class, and then proceed to consider more general classes. Proofs for all results are deferred to Appendix A.

We begin with some basic notation and properties that are useful for games with $n = 2$. Fix $p$, and consider some $H$. For each provider $s_i$, denote the accuracy of its chosen $h_i$ as:

$$a_i = \mathbb{E}_p[\mathbb{1}\{h_i(x) = y\}] \tag{7}$$

---

[2]For completeness, in Appendix B.1 we give the game's characterization as a *congestion game* by identifying the appropriate congestible resources and constructing an explicit cost function.

| | $h_1$ | $h_2$ |
|---|---|---|
| $h_1$ | $\frac{1}{2}a_1, \frac{1}{2}a_1$ | $\frac{1}{2}(a_1 + \delta_{12}), \frac{1}{2}(a_2 + \delta_{21})$ |
| $h_2$ | $\frac{1}{2}(a_2 + \delta_{21}), \frac{1}{2}(a_1 + \delta_{12})$ | $\frac{1}{2}a_2, \frac{1}{2}a_2$ |

Table 1: Payoff matrix of the 2x2 game

We define the *partial discrepancy* of $h_i$ relative to $h_j$ as:

$$\delta_{ij} = \delta_i(h_j) = \mathbb{E}_p[\mathbb{1}\{h_i(x) = y \wedge h_j(x) \neq y\}] \quad (8)$$

which sums points on which $h_i$ is correct on but $h_j$ is wrong.

**Proposition 1.** *Let $h_i, h_j$, then $\mu(h_i \mid h_j) = \frac{1}{2}(a_i + \delta_{ij})$.*

This implies that there are two ways to increase market share: by improving overall accuracy ($a_i$), or by being exclusively correct on more points ($\delta_{ij}$). Thus, the choice of $h$ should consider how these two terms trade off. Note that points in $\delta_{ij}$ are counted twice, since they are also included in $a_i$.

Interestingly, as long as the classifiers are distinct, then whoever has higher accuracy also secures a larger market share:

**Proposition 2.** *For any $h_i \neq h_j$, it holds that:*

$$\mu(h_i \mid h_j) > \mu(h_j \mid h_i) \iff a_i > a_j \quad (9)$$

This, however, should not be taken to imply that maximizing accuracy is a good strategy, since providers seek to maximize their *absolute* market share—not their market share in relation to others. Regardless of the other's market share, a provider may switch to an equally accurate classifier[3]— or even sacrifice in accuracy to gain greater discrepancy— if this results in market share increasing. Empirically we observe that sacrificing accuracy is both common and effective.

### 4.1. Warmup: $2 \times 2$ accuracy markets

Consider a simple setting with $n = 2$ providers and a shared model class of size two, $H = \{h_1, h_2\}$. For example, these could be two available pre-trained models, or an existing model that is already in deployment and a new model that is a possible alternative. Such $2 \times 2$ games are fully determined by the tuple $(a_1, a_2, \delta_{12}, \delta_{21})$—see Table 1. This formulation enables a characterization of all possible equilibria:

**Theorem 1.** *Let $H = \{h_1, h_2\}$. Then for any $p$, the game admits one of two following types:*

1. **Dominant-strategy:** *either $(h_1, h_1)$ or $(h_2, h_2)$ is a PNE*

2. **Anti-coordination:** *both $(h_1, h_2)$ and $(h_2, h_1)$ are PNEs*

In the latter case, the game admits a chicken-like[4] structure: one provider obtains a larger market share by choosing the 'better' classifier, while the other must settle for the smaller share. Note that better here does not mean more accurate, as outcomes at equilibrium also depend on discrepancy. The proof of Thm. 1 relies on the following result:

**Lemma 1.** *Providers will choose differing strategies at equilibrium if and only if $|a_1 - a_2| \leq \frac{1}{3}(\delta_{12} + \delta_{21})$.*

Thus, anti-coordination emerges when $h_1, h_2$ are sufficiently similar in terms of accuracy, but note that the condition is fairly lenient. Empirically, we observe that chicken play is by far the more prevalent scenario, and that the order of play (which is only hinted to here) is highly significant. We next show that the above properties hold more broadly.

### 4.2. Accuracy markets with threshold classifiers

Keeping $n = 2$, consider a more general accuracy game in which $y \in \{0, 1\}$, inputs are scalar ($x \in \mathbb{R}$), and $H$ comprises threshold classifiers $h_\tau(x) = \mathbb{1}\{x > \tau\}$. This also captures settings with general inputs $x$ where there is a pre-trained score function $f(x)$ and each $s_i$ can set its own thresholds as $h_i(x) = \mathbb{1}\{f(x) > \tau_i\}$; i.e., competition revolves around different ways to 'set the bar' w.r.t. $f(x)$.

Our next result shows that under certain conditions, even though $H$ includes a continuum of models, the game simplifies significantly. Consider the following common property:

**Definition 1** (MLR). *Let $p(x, y)$ be continuous in $x$, and $f_y$ the PDF of each conditional $p(x|y)$ for $y \in \{0, 1\}$. We say that $p$ exhibits a (strict) monotone likelihood ratio (MLR) if the density ratio $\rho(x) = \frac{f_1(x)}{f_0(x)}$ is (strictly) increasing.*

We show that MLR entails a simple closed-form solution to the best-response classifier against any other classifier:

**Theorem 2.** *Fix $n = 2$, and let $H = \{h_\tau\}$ be a class of threshold classifiers over $d = 1$. Let $p$ be such that it is strictly MLR in some interval $[a, b]$. Then for any $\tau \in [a, b]$:*

$$\mathrm{BR}_{[a,b]}(\tau) \in \{\max\{a, \rho^{-1}(1/2)\}, \min\{b, \rho^{-1}(2)\}\}$$

*where $\mathrm{BR}_{[a,b]}(\tau)$ is the best-response to $\tau$ from the set $[a, b]$.*

Thm. 2 states that of all thresholds in the range where MLR holds, the set of candidates for a best-response classifier reduces to just two. For natural cases in which the extreme choices of $\tau < a$ and $\tau > b$ are not optimal,[5] the structure of the game simplifies even further.

---

[3]This is a likely scenario: see works on the 'Rashomon' effect (Paes et al., 2023) and model multiplicity (Semenova et al., 2022).

[4]The standard Chicken game has payoffs of the form **T**emptation>**C**oordination>**N**eutral>**P**unishment; our variant has **T**>**N**>**P**>**C**> 0, which preserves the same PNE.

[5]For example, a mixture of two Gaussians is MLR everywhere except possibly in the far tails, where thresholding is ineffective.

**Corollary 1.** *For $n = 2$ and $H = \{h_\tau\}$, any accuracy game played on an MLR region of $p$ reduces to a $2 \times 2$ game. Hence, all results from Sec. 4.1 hold.*

Note the reduced model class is $H = \{\rho^{-1}(1/2), \rho^{-1}(2)\}$.[6] This is not by chance: under MLR, these ratios are precisely the points in which accuracy and discrepancy balance each other. Interestingly, this partitions the population into three segments: mostly negative points, mostly positive, and a mixed subpopulation. Providers then compete over who obtains exclusive access to the more rewarding segments.

Thm. 2 can be generalized to any 1D distribution:

**Theorem 3.** *Fix $n = 2$, and let $H = \{h_\tau\}$ be a class of threshold classifiers. Then for any interval $[a, b]$ and any $\tau$:*

$$\mathrm{BR}_{[a,b]}(\tau) \in \{a, b, \tau\} \cup P_+^{-1}(1/2) \cup P_+^{-1}(2)$$

*where $P_+^{-1}(z) = \{\tau : \rho(\tau) = z \wedge \rho'(\tau) > 0\}$.*

For reasonable distributions, it is likely that $|P_+^{-1}(z)| < c$ for some small constant $c$. This then implies that $H$ effectively reduces to include only $O(c)$ candidate strategies.

The above results also have implications on dynamics:

**Proposition 3.** *For $n = 2$ and $H = \{h_\tau\}$, best-response dynamics converge after one round.*

We therefore receive that for threshold classifiers, not only is calculating the best-response a simple task, but the market also converges immediately. The difference from the MLR setting is that there can now be multiple equilibria, and convergence can depend on the initial choices of $\{h_i^0\}$.

Best-response dynamics imply that model updates can improve market share only for the provider that responds. Interestingly, in the above setting, we can show that a best-response by one player improves outcomes also for the other.

**Proposition 4.** *Let $h_i^0 = h_{opt}$, $\forall i$. Then for each $s_i$, market share $\mu_i$ increases even when the other $s_j$ best-responds.*

Empirically, we observe that this form of implicit cooperation emerges also in broader settings. For $n > 2$, outcomes improve once all other players have responded.

### 4.3. Accuracy markets for general model classes

For general classes and $n = 2$, several properties of the market can still be established. The first considers providers:

**Proposition 5.** *If $h_i^0 = h^0$ $\forall i$, for some $h^0$, then $\mu_i^* \geq \mu_i^0$.*

That is, if providers start at the same initial classifier, then all of them will provably gain from competition. This directly implies that competition also improves welfare for users:

**Corollary 2.** *Fix initial classifier $h^0$ $\forall i$, and let $h^*$ be the set of classifiers at equilibrium. Then $W(h^*) \geq W(h^0)$.*

In terms of the market, we can bound its concentration:

**Proposition 6.** *Let $h^* = (h_i, h_j)$ be any equilibrium. Then $\mu(h_i|h_j) \leq 2 \cdot \mu(h_j|h_i)$.*

Thus, despite the tendency for differentiation under competition, no player can dominate more than $2/3$ of the market.

**General accuracy markets.** Empirically, many of our above results hold also for any number of players and general model classes: quick convergence, implicit cooperation, an incentive to differentiate, sacrificing accuracy for market share, bounded market concentration, and high welfare. Some results however do not carry over to $n > 2$; for example, the order of play becomes much more intricate, and whether moving first is good or bad can depend on context. Importantly, our results hold despite the intractability of computing best responses exactly, and by using our method for learning approximate best responses—presented next.

## 5. Method

We now turn to the question of how to implement a best response, i.e., by solving Eq. (5) for any setting. Our main observation is that a provider's market share objective (Eq. (3)) can be rewritten as a weighted expected accuracy objective with a particular choice of per-example weights:

$$\mu_i = \mathbb{E}_p[w_i(x) \cdot \mathbb{1}\{h_i(x) = y\}], \quad w_i(x) = \frac{1}{1 + \kappa_{-i}(x)} \tag{10}$$

where $\kappa_{-i}(x) = |\{s_j \neq s_i : h_j(x) = y\}|$, i.e., the number of other providers that are correct on $x$. Thus, weights $w_i(x)$ determine the importance of input $x$ for provider $i$, and inform the objective of which inputs to target, or avoid.

**Hardness.** In terms of tractability, our results are mixed:

**Observation 1.** *For any choice of $H$, and given $w_i$, computing the best-response classifier $h_i = \mathrm{BR}(h_{-i})$ is just as hard as maximizing expected accuracy over $H$.*

This holds since weights in Eq. (10) simply modify the data distribution, a change to which learning algorithms should be agnostic.[7] Unfortunately, because maximizing the expected 0-1 accuracy is computationally intractable (and statistically challenging) for the vast majority of classification problems, computing a best-response classifier exactly will mostly be infeasible. The bright side is that this is precisely the problem that machine learning practice aims to solve.

---

[6]If they exist; otherwise the optimal models are at $a, b$; see Thm. 2.

[7]E.g., consider how the PAC framework defines a class $H$ as learnable if there exists an (efficient) algorithm that maximizes accuracy well simultaneously for *all* distributions (Valiant, 1984).

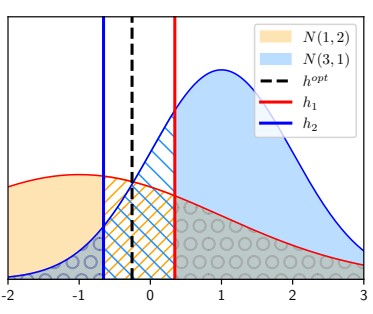 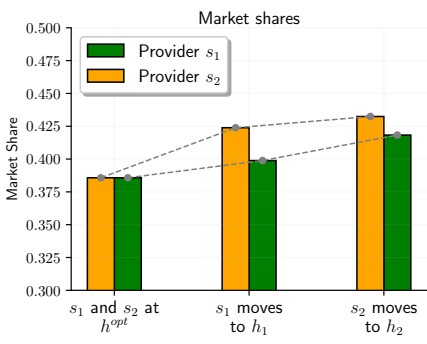 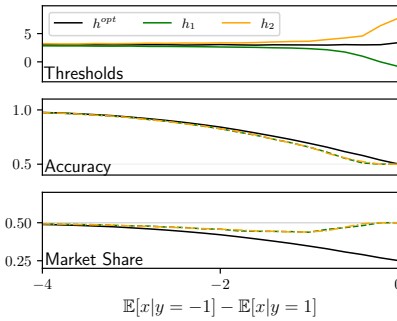

Figure 1: **Two-player threshold market. (Left:)** Data consists of two class-conditional Gaussians $p(x \mid y) = \mathcal{N}(ay, \sigma_y)$ Beginning at $h^{\text{opt}}$, providers compete over who gets the better classifier, $h_2$, which secures exclusive access to the larger sector of positive users (blue). **(Center:)** The game as played over time. Each best response improves market share for both providers, but the second mover ($s_2$) prevails. **(Right:)** Outcomes for increasingly distanced $p(x \mid y)$ (here $\sigma_y = \sigma$). Equilibrium classifiers are pulled further away, and sacrifice accuracy for increased market share.

Hence, any solution that works well for general machine learning tasks should also work well to learn best responses.

**Learning (approximate) best-response classifiers.** Since the game is played on the empirical distribution, we can adapt any method of empirical risk minimization that supports custom example weights, i.e., that aims to solve:

$$\hat{h}_i = \underset{h \in H}{\operatorname{argmin}} \frac{1}{m} \sum_{j=1}^{m} w_i(x_j) \ell(y_j, h(x_j)) + \lambda R(h) \quad (11)$$

where $\ell$ is a proxy loss (e.g., hinge loss or cross-entropy) and $R$ is an (optional) regularization term. This approach applies to any type of data and choice of model class $H$. We refer to $\hat{h}_i$ as the approximate best-response classifier of $s_i$. Note that optimizing $\hat{h}_i$ depends on the other classifiers $h_{-i}$ only through the example weights $w_i(x_j)$. This means that computing the empirical best-response classifier requires only access to the number of other providers that are correct on each data point—not to the actual classifiers in $h_{-i}$. Also, we can observe that the average weight $\frac{1}{|S|} \sum_{i \in S} w_i$ has an intuitive meaning: it is the maximum possible market share that could be achieved by a perfect classifier, i.e., when all its predictions are correct. The actual $\mu_i$ is then this optimal market share minus the weighted loss incurred by the chosen classifier $\hat{h}_i$.

**Performativity.** Eq. (4) casts maximizing market share as a problem of learning under distribution shift, where shift is due to competition, as expressed by weights $w_i(x)$. From the perspective of a single provider $s_i$ at time $t$, weights $w_i^t(x)$ describe how the market has changed *in response to its own actions*, i.e., the choice of $h_i^{t-1}$. This reveals that learning in a market setting is of a performative nature: the choice of classifier at the current time step $t$ shapes the (effective) distribution at the next, $p_i^{t+1}(x, y)$. When there

are only two providers, performativity is *stateless*, meaning that $p_i^{t+1}$ depends only on the current $h_i^t$ through how $s_j$ will best-respond; this relates to the common (and simpler) setting often studied in performative prediction (Perdomo et al., 2020). When $n > 2$, dynamics become *stateful*, i.e., are path-dependent, and so choices accumulate over time—a generally much more challenging setting (Brown et al., 2022; Li & Wai, 2022). Luckily, our market construction adds structure that makes it a tractable instance. An interesting point to make is that performativity in our setting has no 'real' effect on the distribution. Rather, it only changes how providers should *perceive* the distribution in order to effectively maximize their utility in the market.

## 6. Experiments

We now present our empirical investigation of learning in accuracy markets. We begin by demonstrating the basic mechanics of competitive learning on simple synthetic data which allows us to compute best-response classifiers exactly. Then we switch to real data and apply our method from Sec. 5 to accuracy markets across multiple datasets and various learning algorithms. Code is publicly available at https://github.com/BML-Technion/market4acc.

### 6.1. Synthetic data

To gain an understanding of how accuracy markets work, consider a simple setting with $n = 2$ providers, binary labels $y \in \{\pm 1\}$, univariate features $x \in \mathbb{R}$ sampled from class-conditional Gaussians $x \sim p(x \mid y) = \mathcal{N}(ay, \sigma_y)$, and threshold classifiers $H = \{h_\tau(x) = \mathbb{1}\{x > \tau\} \mid \tau \in \mathbb{R}\}$. Figure 1 (left) illustrates this setup for $a = 1, \sigma_{-1} = 2$, and $\sigma_{+1} = 1$, and shows the learned classifiers at equilibrium, $h_1$ and $h_2$; as Thm. 2 suggests, these are precisely $\rho^{-1}(1/2)$ and $\rho^{-1}(2)$. Note how the region between $h_1$ and

Table 2: **Learning in accuracy markets.** Results show outcomes of best-response dynamics, implemented as training by Eq. (11). All runs were initialized to $h^0 = h^{\text{opt}}$, and converged after at most $t = 2$ rounds. Results include % increase at equilibrium of market share (min and max over providers), market concentration (HHI $= \sum_i \mu_i^2$), and welfare. Standard errors of the experiments are insignificant and shown in Appx. D.1.

| | | COMPAS-arrest | | | COMPAS-violent | | | | Adult | | | |
|---|---|---|---|---|---|---|---|---|---|---|---|---|
| | | $\min \mu$ | $\max \mu$ | HHI | welfare | $\min \mu$ | $\max \mu$ | HHI | welfare | $\min \mu$ | $\max \mu$ | HHI | welfare |
| # providers | 2 | +25.4% | +31.0% | +64.5% | +28.2% | +41.8% | +61.2% | +130.6% | +51.5% | +4.0% | +30.5% | +39.8% | +17.3% |
| | 3 | +38.7% | +48.5% | +106.2% | +43.5% | +46.8% | +82.5% | +163.7% | +61.6% | +2.9% | +41.7% | +51.8% | +22.1% |
| | 4 | +38.0% | +66.9% | +121.9% | +48.4% | +48.5% | +94.6% | +172.0% | +63.9% | +6.1% | +39.4% | +51.2% | +22.0% |
| | 5 | +36.8% | +83.2% | +128.7% | +50.1% | +48.7% | +94.6% | +172.3% | +64.0% | +11.7% | +33.2% | +51.4% | +22.7% |
| | 6 | +38.7% | +80.4% | +127.9% | +50.2% | +48.5% | +89.0% | +172.3% | +64.2% | +10.5% | +41.6% | +53.7% | +23.2% |

$h_2$ (hatches) is split between the providers: $s_1$ is exclusively correct on positive examples, and $s_2$ on negatives. The regions to the right of $h_1$ and left of $h_2$ are shared.

Fig. 1 (center) shows how market shares $\mu_1, \mu_2$ evolve over rounds of best-responses. Here we initialize $h_1^0 = h_2^0 = h^{\text{opt}}$ where $h^{\text{opt}}$ is the optimal classifier (i.e., which maximizes accuracy on $p$). In line with our results from Sec. 4.2, dynamics converge after one round, i.e., each provider responds once, and so $h_1 = h_1^1$ and $h_2 = h_2^1$. Although $s_1$ moves first and improves $\mu_1$, this not only improves $\mu_2$ for $s_2$, but also *to a greater extent than that of $s_1$*. When $s_2$ then moves, again both $\mu_1, \mu_2$ increase, but $\mu_2$ retains its advantage over $\mu_1$. Hence, $s_2$ 'wins' the chicken game by playing second and obtaining access to the larger exclusive subgroup of positives. Regardless of who wins, users gain from the competition since welfare ($= \mu_1 + \mu_2$) always increases.

Fig. 1 (right) shows how outcomes change for matching gaussians ($\sigma_{+1}, \sigma_{-1} = 1$) when the class-conditional distributions $p(x \,|\, y = -1)$ and $p(x \,|\, y = 1)$ are pulled closer together, achieved by decreasing $a$. When the distributions are far away, $h_1$ and $h_2$ are at $h^{\text{opt}}$ and so fully share the market. But as overlap increases, several effects take place. First, $h_1$ and $h_2$ grow further apart and become more distinct in who they target, causing the exclusivity regions to grow in size. Second, since classification becomes harder, the maximal attainable accuracy decreases. The accuracies of $h_1, h_2$ also decrease, but at a faster rate—a result of specialization. Third, we see that welfare—as the sum of market shares—begins at its maximal value of 1 (when the gaussians are perfectly separable), then decreases when the exclusivity doesn't extend to the tails, and finally returns to 1 when the providers play opposite thresholds. We explore additional aspects on non-matching gaussians in Appendix D.4.

### 6.2. Real data

Our goal in this section is to explore accuracy markets under our three perspectives: (learning) providers, users, and the market. We experiment with three datasets: COMPAS-Arrest, COMPAS-Violence, and Adult, and consider several learning algorithms, including linear SVMs, boosted trees (using XGBoost), and random forests. These generally work well for standard accuracy tasks on the above datasets. Appendix C includes full details on datasets, methods, and our experimental setup. Appendix D includes additional experiments that extend and complement those presented here.

**Learning.** Table 2 shows performance under several measures of interest across multiple datasets and for varying number of providers. Here we show results for Linear SVMs, but note that other learning algorithms exhibit overall similar trends (see Appendix D.1). All results are averaged over 10 random train-test splits. The table describes outcomes after $t = 2$ rounds, which we found sufficed to obtain near-convergence across all settings—regardless of training set size, model class complexity, and the use of a proxy objective to implement (approximate) best responses.

In terms of market share improvement, we see that competition is helpful for all providers; nonetheless, there can be a large gap between the minimal and maximal improvement. In the COMPAS datasets, this gap begins at smaller values, but grows as $n$ increases. For Adult, whose baseline accuracy is higher, the gap remains mostly stable, but is large to begin with. As competition progresses, the market becomes more concentrated, which also generally increases with $n$. Overall welfare gains are quite high, reaching up to $+65\%$.

**Market outcomes.** Figure 2 shows how the market is partitioned across providers at equilibrium. Here we focus on COMPAS-Arrest with XGBoost and $n = 3$ providers; providers are numbered by their order of play (i.e., $s_1 \prec s_2 \prec s_3$), where this order is preserved across rounds. The plot shows for each provider $s_i$ its total market share (bottom left) and accuracy on the entire population (top left) due to its final learned $h_i$. The plot also shows the decomposition of the market across all subsets of providers: what proportion is exclusive to $s_1$, what is joint to $s_2$ and $s_3$, what is shared by all, etc. (right). Here we see that $s_1$, who moved first, attained the largest market share (36%). However, its accuracy is significantly lower than others,

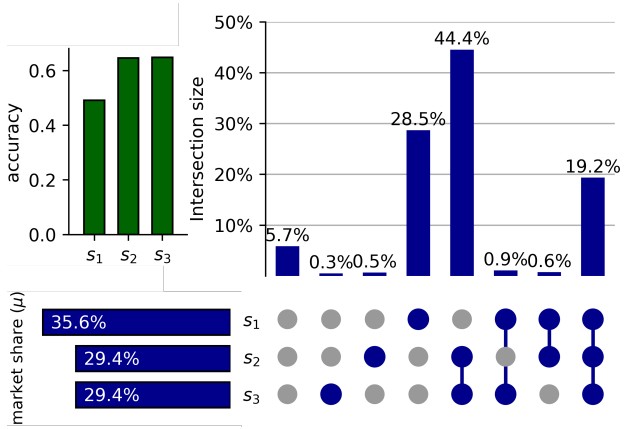

Figure 2: **Market share.** An example for $n = 3$ where the first mover ($s_1$) dominates the market, achieved by sacrificing overall accuracy for exclusive access to a large user sector. Other providers are left to share the remaining sector.

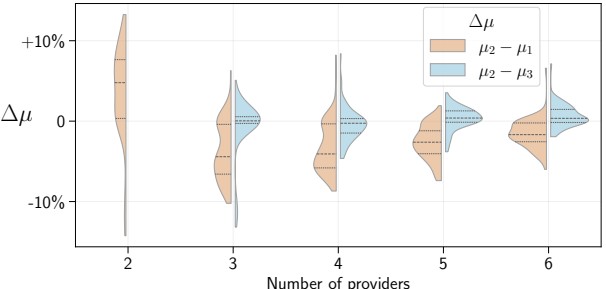

Figure 3: **Influence of order of play on market share.** The orange densities measure the difference in market share between the provider that moved 2nd and the provider that moved 1st, and the blue densities measure the difference in market share between the provider that moved 2nd and the provider that moved 3rd, for $n > 2$. Dashed lines inside the densities represent the 25%, 50%, and 75% quantiles of values, respectively, from bottom to top.

and below 50%. The subsets plot reveals the reason: $s_1$ was able to gain exclusive access to 28.5% of the market; it shares an additional 19% with all providers, but only 1% with each of them alone. In contrast, $s_2$ and $s_3$ share almost all of their users, either as a pair (44%) or along with $s_1$. This shows how $s_1$ has come to dominate the market by learning a classifier $h_1$ that sacrifices accuracy in order to effectively target an exclusive user sector. The low overall accuracy of $h_1$ suggests that naïvely optimizing for accuracy without considering the effects of competition can be highly suboptimal in terms of market outcomes.

**Order of play.** Whereas our $2 \times 2$ analysis from Sec. 4.1 suggested that the game either has a dominant strategy or a chicken-like structure (which implies that moving second is preferable), we see in Fig. 2 that for $n > 2$ providers reality is more complex, and in fact moving *first* allows to dominate the market. Interestingly, this first move induces a 2-player game on the other providers whose equilibrium admits a dominant strategy. To quantify this phenomenon, and assert its robustness across methods, we ran experiments for every combination of datasets and model classes, listed in Appendix C. For each experiment, namely for each combination of model class and dataset, the final market shares were calculated for each provider along with his/her relative position of play, i.e., at what position did the provider perform a best-response. Additionally, each experiment was performed for competitions with 2,3,4,5, and 6 players, so that we can compare dynamics across different market saturations. Figure 3 shows the order of play comparisons, averaged out across all of the experiments that were described above. When the competition game is played with $n = 2$ providers, we see a clear preference to be the provider that moves last, characterized by the market share term $\mu_2$.

The vast majority of experiments showed a significant gain in market share, as seen by the fact that the 25% quantile of values already shows a net-positive gain from moving last. We also note that the expected (mean) competitive advantage of moving last is 3.6% with a median of 4.8%. Given that for 2 players with equal model classes, the market share of a single provider will never exceed $\frac{2}{3} = 0.67$ (see Proposition 6), then a 4.8% difference in market share is quite significant.

When the competition game is played with $n \geq 3$ providers, however, we observe an entirely different dynamic. Figure 3 provides two densities: The orange density is as in the 2-provider setting ($\mu_2 - \mu_1$). The blue density is the difference in market share between the player who moved 2nd versus the player who moved 3rd ($\mu_2 - \mu_3$). We find here that the ratios have switched: it is in fact more advantageous to be the provider that moves 1st, as evidenced by the negative orientation of the orange density plot. The next interesting thing that we note is the seeming insignificance of order-of-play beyond the first two positions, as we can observe that the blue density hovers around 0 with relatively low variance. This alludes to the premise put stated above that in competitive settings with 3 or more players, the person who moves 1st is in essence "grabbing his territory", which then induces all of the other players to play among themselves for the other resources, i.e, consumers.

**User welfare.** Our result in Cor. 2 states that competition is conductive to welfare. It remains to consider *how* conductive it is, as well as how *quickly* welfare improves. Fig. 4 left shows how welfare changes over time and for increasing number of providers $n$. Here we focus on COMPAS-arrest and LinearSVC, with other settings shown in Appendix D.3.

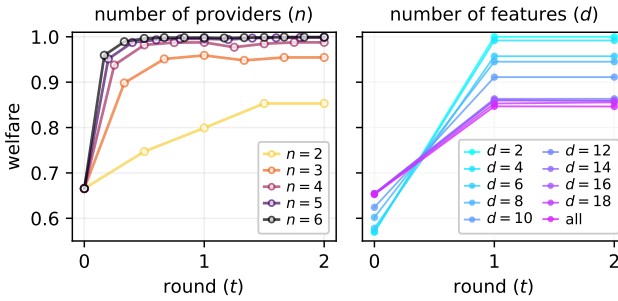

Figure 4: **Welfare over rounds.** Competition consistently increases welfare over time, until convergence. Welfare improves with the number of providers **(left)**, but (perhaps counterintuitively) decreases with the quality of data **(right)**.

The plot shows a clear trend of welfare increasing throughout competition. It also makes apparent the effect of $n$: as the number of providers increases, welfare climbs higher and faster. For $n = 2$, welfare attains a maximum of 0.85, reached only at the second round. For $n = 3$, welfare maximizes at 0.95 by the end of the first round. Notice that welfare reaches the upper bound of 1 already at $n = 5$ and before the end of the first round (i.e., before all providers have moved). With $n = 6$ providers, this occurs even earlier.

The above depicts accuracy markets as highly efficient. On the one hand, our market setting allows for the free flow of information, and models users as making informed (rational) decisions—which are necessary to enable efficient outcomes. But on the other, providers are restricted in that they cannot compute best-responses exactly. We therefore take the results above to again suggest that maximizing market share using proxy objectives can work well in practice.

Since market share should generally align with accuracy, another interesting question is how does the capacity to maximize accuracy affect the overall welfare. For a different setting of competing predictors, Jagadeesan et al. (2024) argue that increased capacity can result in *lower* welfare for users. We show that this occurs quite distinctly in our setting as well: When the complexity goes down, maximizing accuracy may be harder, but gaining discrepancy is easier, as there are more users to specialize on. Following the idea of controlling capacity by the quality of representation, we implement this by varying the number of features available for learning. Fig. 4 (right) shows welfare for increasing number of features and for $n = 2$ (see Appendix D.3 for more settings). As expected, at time $t = 0$, better representations entail higher accuracy, and therefore higher welfare. But once providers respond, the trend inverts: restricting learning to use only two features attains the optimal welfare of 1, while using all features gives welfare of $0.84$.

## 7. Discussion

This work studies learning in a competitive setting where classifiers are trained to increase market share. From a learning perspective, our main message is that while maximizing accuracy naïvely is likely not a good strategy, optimizing a weighted accuracy objective that correctly encodes competition can be very effective. Although technically similar, the transition to market-induced objectives has implications on the market and consequently on user welfare. In our market model competition promotes welfare, but this relies on model transparency, efficient information flow, and calculated user decisions. Realistic markets are likely to fall short of such ideals: firms may prefer to keep models private, informational advantages can be exploited, and user behavior can be far from rational. The fact that most service sectors currently include only a few competing platforms (consider media, social, e-commerce, finance, housing, etc.) should raise concerns of oligopolistic behavior, notably collusion and lock-in practices. This requires deliberation of appropriate regulation for these emerging accuracy markets.

## Impact Statement

Our work considers the role of machine learning in fostering markets in which utility to consumers derives from personalized accuracy. The market model we study is inspired by real markets of this type, but as a model, makes several simplifying assumptions that merit consideration before drawing conclusions about actual markets. One assumption is that there is a single underlying distribution that is fixed and accessible to all providers. Although this is a common assumption in standard machine learning, under competition it has further implications. When data distributions differ across providers, or even when the distribution is the same but samples are different (which is plausible), it is no longer clear if pure equilibrium exists or is reachable through reasonable dynamics. Another assumption is that users choose a provider who is accurate on their own input. This applies in some cases, such as personalized recommendations: users likely know their preferences, but do not know if (and which) content items match them—but can make conclusions once recommended. More generally however, we consider our model as a simplification of outcomes that materialize and stabilize over time, such as provider reputation, social learning, or confirmation in hindsight. Another alternative is that users interact with a platform not once but many times; if the platform has access to some personalized examples, then competition can revolve around future expected outcomes. A final assumption is that users are rational and choose by maximizing utility independently at each time step. This can be a reasonable assumption in settings where users have both incentive and resources to invest effort in bettering their choices, and when sufficient

time passes between rounds to enable switching providers. More generally, user behavior is likely to play a key role in market outcomes, and can benefit from more realistic modeling. It is also likely that competition can drive providers to exploit users' behavioral weaknesses—an additional reason for establishing appropriate regulations and norms.

## Acknowledgments

The authors would like to thank Fan Yao, Moran Koren, Omer Ben-Porat, and Eden Saig for their insightful remarks and valuable suggestions. This work is supported by the Israel Science Foundation grant no. 278/22.

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

# A. Proofs

**Notations.** In the proofs below, we may use the multiple notations for *partial discrepancy* interchangeably, namely $\delta_{ij} = \delta_{h_i}(h_j)$. This is for readability and ease of portrayal. Similarly, in place of $\mu$ (market share), sometimes there can be written $U$, for utility. These terms are also interchangeable. Lastly, the terms "player" and "provider" are interchangeable as well, since both refer to the learner.

Before delving in to the proofs of the individual statements from the main paper, we will start with showing a helpful observation that sits at the crux of competitive dynamics, and will help with many of the proofs below.

**Observation 2.** *For any 2 classifiers $h_i, h_j$ with accuracies $a_i, a_j$, it stands: $a_i - \delta_{ij} = a_j - \delta_{ji}$.*

*Proof.* Visual proof by building a confusion matrix split up by accuracy on the label:

|  | $h_j = y$ | $h_j \neq y$ |  |
|---|---|---|---|
| $h_i = y$ | $a_i - \delta_{ij}$ | $\delta_{ij}$ | $= a_i$ |
| $h_i \neq y$ | $\delta_{ji}$ | $1 - a_i - \delta_{ji}$ | $= 1 - a_i$ |
|  | $= a_j$ | $= 1 - a_j$ |  |

Explanation of the table:

- the cells are partitioned by buckets according to the correctness of the classsifiers. This helps us understand the discrepancy and overlap of accurate predictions between the classifiers.

- We can immmediately observe that the 1st row sums to $a_i$, and the 1st column sums to $a_j$. Similarly, the 2nd row and 2nd column sum up to $1 - a_i, 1 - a_j$, repsectively. So too, $d_{ij}$ and $\delta_{ji}$ are in the top right and bottom left cell by definition.

From the sums of the rows and columns, we observe the following equality from the top-left cell:

$$a_i - \delta_{ij} = a_j - \delta_{ji} \tag{12}$$

$\square$

**Proposition 1 (Market Share).**

*Proof.* from the definition of our problem setting, classifier $h_j$ gets:

- Full market share on the points that only $h_j$ is correct on: $\frac{1}{n}\sum_{i=1}^n \mathbb{1}\{h_j(x) = y_i \wedge h_i(x) \neq y_i\}$

- Half market share on the points that they are both correct on: $\frac{1}{2} \cdot \frac{1}{n}\sum_{i=1}^n \mathbb{1}\{h_j(x) = y_i \wedge h_i(x) = y_i\}$

By the definition of partial discrepancy (Eq. 8), the first bullet is exactly $\delta_{h_j}(h_i)$, and the second bullet is equal to $\frac{1}{2}(a_j - \delta_{h_j}(h_i))$.

We then receive: $U(h_j|h_i) = \delta_{h_j}(h_i) + \frac{1}{2}(a_j - \delta_{h_j}(h_i)) = \frac{1}{2}(a_j + \delta_{h_j}(h_i))$ $\square$

**Proposition 2 (Market share ↔ accuracy).**    We will first prove a helpful claim:

*Helpful claim* 1.  For any 2 classifiers $h_1, h_2$ i with accuracies $a_1, a_2$, respectively, it stands that $\delta_{12} > \delta_{21} \Leftrightarrow a_1 > a_2$.

*Proof.*  From Observation 2, $a_1 - \delta_{12} = a_2 - \delta_{21}$, meaning: $a_1 = a_2 + \delta_{12} - \delta_{21}$. Therefore:

$$
\begin{aligned}
a_1 > a_2 &\Leftrightarrow a_1 - a_2 \geq 0 \\
&\Leftrightarrow a_2 + \delta_{12} - \delta_{21} - a_2 \geq 0 \\
&\Leftrightarrow \delta_{12} - \delta_{21} \geq 0 \\
&\Leftrightarrow \delta_{12} \geq \delta_{21}
\end{aligned}
$$

Where the 2nd inequality comes from substituting $a_1 = a_2 + \delta_{12} - \delta_{21}$  □

We will now prove Proposition 2:

*Proof.*  From Proposition 1, We know: $\mu(h_1|h_2) = a_1 + \delta_{12}$, and $\mu(h_2|h_1) = a_2 + \delta_{21}$.

$$
\begin{aligned}
&\mu(h_1 \mid h_2) > \mu(h_2 \mid h_1) \\
\Leftrightarrow \quad & a_1 + \delta_{12} > a_2 + \delta_{21} \\
\Leftrightarrow \quad & a_1 > a_2
\end{aligned}
\tag{13}
$$

Where the last inequality follows from Helpful claim 1.  □

**Lemma 1.**    We will start with a helpful claim:

*Helpful claim* 2.  Let $h_1, h_2 \in \mathcal{H}$. Then:

- $U(h_1|h_2) > U(h_2|h_2) \Leftrightarrow \delta_{h_1} > \frac{1}{2}\delta_{h_2}$

- $U(h_2|h_1) > U(h_1|h_1) \Leftrightarrow \delta_{h_2} > \frac{1}{2}\delta_{h_1}$

Additionally, the above inequalities hold **if and only if** $|\delta_{h_1} - \delta_{h_2}| < \frac{1}{3}(\delta_{12} + \delta_{21})$.

*Proof.*  From Proposition 1 we know that $U(h_1|h_1) = \frac{1}{2}a_1, U(h_2|h_2) = \frac{1}{2}a_2, U(h_i|h_j) = \frac{1}{2}(a_i + \delta_{ij})$. Therefore,

$$
\begin{aligned}
U(h_1|h_2) > U(h_2|h_2) &\Leftrightarrow \frac{1}{2}(a_1 + \delta_{12}) > \frac{1}{2}(a_2) \\
&\Leftrightarrow a_1 + \delta_{h_1} > a_1 + \delta_{h_2} - \delta_{h_1} \\
&\Leftrightarrow \delta_{h_1} > \delta_{h_2} - \delta_{h_1} \\
&\Leftrightarrow 2\delta_{h_1} > \delta_{h_2} \\
&\Leftrightarrow \delta_{h_1} > \frac{1}{2}\delta_{h_2}
\end{aligned}
\tag{14}
$$

Where the substitution of the RHS in the 2nd line comes from Observation 2.

Similarly,

$$
\begin{aligned}
U(h_2|h_1) > U(h_1|h_1) &\Leftrightarrow a_2 + \delta_{h_2} > a_1 \\
&\Leftrightarrow a_2 + \delta_{h_2} > a_2 + \delta_{h_1} - \delta_{h_2} \\
&\Leftrightarrow \delta_{h_2} > \delta_{h_1} - \delta_{h_2} \\
&\Leftrightarrow 2\delta_{h_2} > \delta_{h_1} \\
&\Leftrightarrow \delta_{h_2} > \frac{1}{2}\delta_{h_1}
\end{aligned}
\tag{15}
$$

Now assume that the inequalities hold, meaning $\delta_{h_1} > \frac{1}{2}\delta_{h_2}$ and $\delta_{h_2} > \frac{1}{2}\delta_{h_1}$. Then,

$$\delta_{h_1} > \frac{1}{2}\delta_{h_2} \rightarrow \delta_{h_1} + \delta_{h_2} > \frac{3}{2}\delta_{h_2}$$

$$\rightarrow \delta_{h_2} < \frac{2}{3}(\delta_{h_1} + \delta_{h_2}) \tag{16}$$

Similarly, $\delta_{h_1} < \frac{2}{3}(\delta_{h_1} + \delta_{h_2})$. This means that $\max\{\delta_{h_1}, \delta_{h_2}\} < \frac{2}{3}(\delta_{h_1} + \delta_{h_2})$, and therefore $|\delta_{h_1} - \delta_{h_2}| < \frac{1}{3}(\delta_{h_1} + \delta_{h_2})$.

[Note that all derivations apply both ways, meaning if $|\delta_{h_1} - \delta_{h_2}| < \frac{1}{3}(\delta_{h_1} + \delta_{h_2})$, then $\delta_{h_1} > \frac{1}{2}\delta_{h_2}$ and $\delta_{h_2} > \frac{1}{2}\delta_{h_1}$.]

$\square$

Using Helpful claim 2, the proof of Lemma 1 is almost immediate: from Observation 2 we know that $a_1 - a_2 = \delta_{12} - \delta_{21}$.

The 2 players will choose differing strategies in the 2x2 game if and only if $U(h_1|h_2) > U(h_2|h_2)$ **and** $U(h_2|h_1) > U(h_1|h_1)$, which holds if and only if $\delta_{h_1} > \frac{1}{2}\delta_{h_2}$ **and** $\delta_{h_2} > \frac{1}{2}\delta_{h_1}$, which holds if and only if $|\delta_{h_1} - \delta_{h_2}| < \frac{1}{3}(\delta_{12} + \delta_{21})$. Since we know $a_1 - a_2 = \delta_{12} - \delta_{21}$, this proves Lemma 1.

**Theorem 1 (2x2 PNEs.)** From the inequalities in Lemma 1, we receive the conditions for which the providers play anti-coordinated strategies.

If one of these inequalities doesn't hold, meaning either $U(h_1|h_2) < U(h_2|h_2)$ or $U(h_2|h_1) < U(h_1|h_1)$, then the dominant strategy is $h_1, h_2$, respectively, depending on which inequality does not hold.

**Theorem 2 (Threshold best-responses under MLR).**

*Proof.* Firstly, we note that since $g$ is continuous[8] and increasing strongly in $[a, b]$, $g^{-1}$ is well defined.

We will split $[a, b]$ into sub-intervals $[a, h]$, $[h, b]$ and calculate the best response in each interval:

Let $h$ be the strategy we are responding to.

It is clear that for all strategies in $[a, h]$, the utility on all points in $[h, b]$ is constant, since the classification on those points is the same. So within the interval $[a, h]$, the player is looking to maximize $U_{[a,h]}$. Similarly, when considering the best response in interval $[h, b]$, we need only maximize $U_{[h,b]}$, since for all points in the interval $U_{[a,h]}$ is the same.

Let $U_{[a,b]}(h|h) = c$.

It stands that $\forall h_1 \in [a, h]$:

$$U_{[a,b]}(h_1|h) - c = \int_{h_1}^{h} f_1(x) - \frac{1}{2}f_0(x)dx$$

This is a straightforward expression of the utility. Any points outside $[h_1, h]$ have an identical classification for both $h$ and $h_1$, and so the only difference in utility is found inside the interval $[h_1, h]$, which $h_1$ classifies as positive and $h$ classifies as negative. Therefore the gain in utility is $\int_{h_1}^{h} f_1(x)dx$, but the loss on the negative points is $\frac{1}{2}\int_{h_1}^{h} f_0(x)dx$, since the utility on those points would be otherwise shared with $h$.

Similarly, $\forall h_1 \in [h, b]$:

$$U_{[a,b]}(h_1|h) - c = \int_{h}^{h_1} f_0(x) - \frac{1}{2}f_1(x)dx$$

Therefore, $BR_{[a,h]}(h) = \underset{h_1 \in [a,h]}{\operatorname{argmax}} U_{[a,b]}(h_1|h) = \underset{h_1 \in [a,h]}{\operatorname{argmax}} \int_{h_1}^{h} f_1(x) - \frac{1}{2}f_0(x)dx$.

And $BR_{[h,b]}(h) = \underset{h_1 \in [h,b]}{\operatorname{argmax}} U_{[a,b]}(h_1|h) = \underset{h_1 \in [h,b]}{\operatorname{argmax}} \int_{h_1}^{h} f_0(x) - \frac{1}{2}f_1(x)dx$.

We will divide into cases based on the value of $g(h)$:

---

[8] We note that the theorem holds for categorical distributions as well, where the provider will simply go to the nearest point $x : g(x) \geq \frac{1}{2}$ if to the left, or $x : g(x) \leq 2$ if doing a best-response to the right.

**Case 1:** $1/2 \leq g(h) \leq 2$**:**

We will calculate $BR_{[a,h]}(h)$. Since $g$ is strictly increasing in $[a, b]$, the value $f_0(x) - \frac{1}{2}f_1(x)$ is positive for all points $x$ where $g(x) \geq \frac{1}{2}$.

Therefore, $\forall h_1 < h_2 \in [a, h]$, if $g(h_1) \geq \frac{1}{2}$ then $U_{[a,h]}(h_1|h) \geq U_{[a,h]}(h_2|h)$

In this case: $BR_{[a,h]}(h) = \max\left(a, g^{-1}(1/2)\right)$, since in the case where $\forall h_1 \in [a, h], g(h_1) > \frac{1}{2}$, then the maximum utility is found at $a$.

Similarly, the same argument holds to derive that: $BR_{[h,b]}(h) = \max\left(b, g^{-1}(2)\right)$

**Case 2:** $g(h) > 2$**:**

In this case, $\forall h_1 > h$ it stands that $U_{[h,b]}(h_1|h) < U_{[h,b]}(h|h)$, since the value $f_0(x) - \frac{1}{2}f_1(x)$ is negative for all points in $[h, b]$.

Therefore the best-response is found in the interval $[a, h]$, in which case the derivation from the previous case holds, and so $BR_{[a,h]}(h) = \max\left(a, g^{-1}(1/2)\right)$.

**Case 3:** $g(h) < 1/2$**:**

In this case, $\forall h_1 < h$ it stands that $U_{[a,h]}(h_1|h) < U_{[a,h]}(h|h)$, since the value $f_0(x) - \frac{1}{2}f_1(x)$ is negative for all points in $[a, h]$.

Therefore the best-response is found in the interval $[h, b]$, in which case the derivation from the previous case holds, and so $BR_{[h,b]}(h) = \max\left(b, g^{-1}(2)\right)$.

So across all cases, we find that in the interval $[a, b]$, there are only 2 possible best responses:

$\max\left(b, g^{-1}(2)\right)$, and $\min\left(a, g^{-1}(1/2)\right)$.

□

**Corollary 1.**

*Proof.* Immediate from the fact that the strategy space of both players can be reduced to the 2 candidate best-responses stipulated in Theorem 2.

□

**Theorem 3 (General threshold best-responses).**

*Proof.* As in the proof of Theorem 2, We will split $[a, b]$ into sub-intervals $[a, h], [h, b]$ and calculate the set of possible best responses for each interval:

Let $h$ be the strategy/threshold we are responding to. We will rewrite the explicit forms for $U_{[a,b]}$:

$\forall h_1 \in [a, h]$:

$$U_{[a,b]}(h_1|h) - c = \int_{h_1}^{h} f_1(x) - \frac{1}{2}f_0(x)dx$$

$\forall h_1 \in [h, b]$:

$$U_{[a,b]}(h_1|h) - c = \int_{h_1}^{h} f_0(x) - \frac{1}{2}f_1(x)dx$$

Where $c = U_{[a,b]}(h|h)$.

As explained in the proof of Theorem 2, to calculate the best response in $[a, h]$, it is sufficient to maximize $U_{[a,h]}$; and to calculate the best-response in $[h, b]$, it is sufficient to maximize $U_{[h,b]}$.

We will calculate the best response in $[a, h]$:

*Helpful claim* 3. Let $(h_1, h_2)$ be any open interval in $[a, h]$ such that $\forall x \in (h_1, h_2)$ it holds that $g(x) > \frac{1}{2}$.

Then $\forall x \in (h_1, h_2] \rightarrow U_{[a,h]}(h_1|h) > U_{[a,h]}(x|h)$.

*Proof.* Let $x \in (h_1, h_2]$.

Using the derivations above of relative market shares between thresholds, we will compare the market shares of $x$ and $h_1$:

$U_{[a,h]}(h_1|h) - U_{[a,h]}(x|h) = \int_{h_1}^{x} f_1(u) - \frac{1}{2} f_0(u) du.$

Since we are given that in the segment $[h_1, x]$, $g > \frac{1}{2}$, then it folows that $\forall u$, $f_1(u) - \frac{1}{2} f_0(u) > 0$, and therefore the integral must be positive and hence $U_{[a,h]}(h_1|h) - U_{[a,h]}(x|h) > 0$.

$\square$

*Helpful claim* 4. Let $(h_1, h_2)$ be any interval in $[a, h]$ such that $\forall x \in (h_1, h_2)$ it holds that $g(x) < \frac{1}{2}$.

Then $\forall x \in [h_1, h_2) \rightarrow U_{[a,h]}(h_2|h) > U_{[a,h]}(x|h)$.

*Proof.* We will show that $U_{[a,h]}(h_2|h) - U_{[a,h]}(x|h) > 0$, in a similar manner to Helpful claim 3:

Let $x \in (h_1, h_2]$. We will compare the market shares of $x$ and $h_1$:

$U_{[a,h]}(h_2|h) - U_{[a,h]}(x|h) = \int_{x}^{h_1} f_0(u) - \frac{1}{2} f_1(u) du.$

Since we are given that in the segment $[x, h_2]$, $g < \frac{1}{2}$, then it folows that $\forall u$, $f_0(u) - \frac{1}{2} f_1(u) > 0$, and therefore the integral must be positive and hence $U_{[a,h]}(h_2|h) - U_{[a,h]}(x|h) > 0$.

$\square$

Using the helpful claims, we can see that $BR_{[a,h]}(h) \in \{a, h\} \cup P_+^{-1}(1/2)$:

Let $h_1 \notin \{a, h\} \cup P_+^{-1}(1/2)$.

Case 1 - $g(h_1) > \frac{1}{2}$:

Let $(x, y) \subseteq [a, h]$ be the largest consecutive interval that includes $h_1$ such that $\forall h' \in (x, y) \rightarrow g(h') > \frac{1}{2}$. Since $f_0, f_1$ are continuous, then we know $g$ is continuous. Therefore, either $g(x) = \frac{1}{2}$ and $g'(x) > 0$, or $x = a$. From Helpful claim 3, we receive that $U_{[a,h]}(x|h) > U_{[a,h]}(h_1|h)$, and therefore $h_1$ cannot be a best response.

Case 2 - $g(h_1) < \frac{1}{2}$:

Let $(x, y) \subseteq [a, h]$ be the largest consecutive interval that includes $h_1$ such that $\forall h' \in (x, y) \rightarrow g(h') < \frac{1}{2}$. Since $f_0, f_1$ are continuous, then we know $g$ is continuous. Therefore, either $g(y) = \frac{1}{2}$ and $g'(x) > 0$, or $y = h$. From Helpful claim 4, we receive that $U_{[a,h]}(y|h) > U_{[a,h]}(h_1|h)$, and therefore $h_1$ cannot be a best response.

We define similar claims to calculate the set of possible best responses in $[h, b]$:

*Helpful claim* 5. Let $(h_1, h_2)$ be any open interval in $[h, b]$ such that $\forall x \in (h_1, h_2)$ it holds that $g(x) > 2$.

Then $\forall x \in (h_1, h_2] \rightarrow U_{[h,b]}(h_1|h) > U_{[a,h]}(x|h)$.

*Proof.* We will show that $U_{[h,b]}(h_1|h) - U_{[h,b]}(x|h) > 0$:

Let $x \in (h_1, h_2]$. We will compare the market shares of $x$ and $h_1$:

$U_{[h,b]}(h_1|h) - U_{[h,b]}(x|h) = \int_{h_1}^{x} \frac{1}{2} f_1(u) - f_0(u) du.$

Since we are given that in the segment $[x, h_2]$, $g > 2$, then it follows that $\forall u$, $\frac{1}{2} f_1(u) - f_0(u) > 0$, and therefore the integral must be positive and hence $U_{[h,b]}(h_1|h) - U_{[h,b]}(x|h) > 0$. $\square$

*Helpful claim* 6. Let $(h_1, h_2)$ be any open interval in $[h, b]$ such that $\forall x \in (h_1, h_2)$ it holds that $g(x) < 2$.

Then $\forall x \in [h_1, h_2) \rightarrow U_{[a,h]}(h_2|h) > U_{[a,h]}(x|h)$.

*Proof.* We will show that $U_{[h,b]}(h_2|h) - U_{[h,b]}(x|h) > 0$:

Let $x \in (h_1, h_2]$. We will compare the market shares of $x$ and $h_1$:

$U_{[h,b]}(h_2|h) - U_{[h,b]}(x|h) = \int_x^{h_2} f_0(u) - \frac{1}{2}f_1(u)du$.

Since we are given that in the segment $[x, h_2]$, $g < 2$, then it follows that $\forall u, f_0(u) - \frac{1}{2}f_1(u) > 0$, and therefore the integral must be positive and hence $U_{[h,b]}(h_2|h) - U_{[h,b]}(x|h) > 0$.

$\square$

Using the helpful claims, we can see that $BR_{[h,b]} \in \{h, b\} \cup P_+^{-1}(2)$:

Let $h_1 \notin \{b\} \cup P_+^{-1}(2)$.

Case 1 - $g(h_1) > 2$:

Let $(x, y) \subseteq [a, b]$ be the largest consecutive interval that includes $h_1$ such that $\forall h' \in (x, y) \rightarrow g(h') > 2$. Since $f_0, f_1$ are continuous, then we know $g$ is continuous. Therefore, either $g(x) = 2$ and $g'(x) > 0$, or $x = a$. From Helpful claim 5, we receive that $U_{[a,h]}(x|h) > U_{[a,h]}(h_1|h)$, and therefore $h_1$ cannot be a best response.

Case 2 - $g(h_1) < 2$:

Let $(x, y) \subseteq [h, b]$ be the largest consecutive interval that includes $h_1$ such that $\forall h' \in (x, y) \rightarrow g(h') < 2$. Since $f_0, f_1$ are continuous, then we know $g$ is continuous. Therefore, either $g(y) = 2$ and $g'(x) > 0$, or $y = b$. From Helpful claim 5, we receive that $U_{[a,h]}(y|h) > U_{[a,h]}(h_1|h)$, and therefore $h_1$ cannot be a best response.

$\square$

**Proposition 3 (Convergence after 1 round).**

*Proof.* Let $h$ be any starting classifier.

At timestep $t = 0$, we asume both players are at $h$.

At timestep $t = 1$, player i plays $h_i^1 = BR(h)$, and player j plays $h_j^1 = BR(h_1^1)$.

Let $h_{min} = \min\{h_i^1, h_j^1\}, h_{max} = \max\{h_i^1, h_j^1\}$.

Firstly, we will argue that there exists an optimal-accuracy classifier $h_{opt}$ such that $h_{opt} \in [h_{min}, h_{max}]$:

Assume for the sake of contradiction that this isn't the case. Then there must exist some $h_{opt}$ either to the left of $h_{min}$ or to the right of $h_{max}$. Let's assume w.l.o.g that there exists some $h_{opt} > h_{max}$. Then $\mu(h_{opt}|h_{min}) > \mu(h_{max}|h_{min})$: $a_{opt} > a_{max}$ by definition, and $\delta_{opt,min} > \delta_{max,min}$, since when $h_{min} < h_{max} < h_{opt}$, then $\delta_{max,min} \subset \delta_{opt,min}$.[9]

Therefore, exists some $h_{opt} \in [h_{min}, h_{max}]$.

We now argue that $(h_i^1, h_j^1)$ is a PNE.

Assume without loss of generality $h_i^1 < h_{opt}$. This generalization is without loss since we are proving a best-response equilibrium symmetrically for both thresholds, so it does not matter which player is on which side of $h_{opt}$.

From the proof of Theorem 3, we know that $h_i^1 = \underset{h \in \{a, h_{opt}, P_+^{-1}(1/2)\}}{\operatorname{argmax}} U(h|h_{opt})$.

[if $h_i^1 = h_{opt}$ we are done.]

We know then that $h_j^1 \geq h_{opt}$.

Assume for the sake of contradiction that $h_j^1 < h_{opt}$ -

Then $\delta_{h_{opt}}(h_i^1) > \delta_{h_j^1}(h_i^1)$ , and from the optimality of $h_{opt}$: $a_{opt} \geq a_{h_j^1}$, and therefore $U(h_{opt}|h_i^1) > U(h_j^1|h_i^1)$, contradiction to $h_j^1$ being a best-response.

Now, $h_j^1 \geq h_{opt} > h_i^1$.

We will argue $h_i^1$ is a best-response to $h_j^1$:

_______________

[9]containment here refers to the points that contribute to the values of $\delta$

$\forall h \in (h_{opt}, h_j^1]$, the utility of $h_{opt}$ is greater, similarly to how was argued above.

$\forall h < h_{opt}$, if $h_i^1$ is a best-response to $h_{opt}$, then it must also be a best-response to $h_j^1$, since the accuracy stays the same and the discrepancy grows in an equal amount for all classifiers $h < h_{opt}$.

Therefore, both classifiers $h_i^1, h_j^1$ are best responses to each other and therefore are a PNE. □

**Proposition 4 ("I improve, you improve").**

*Proof.* From the proof of convergence in Proposition 3, we receive that in all threshold games, the players go to either side of an optimal classifier $h_{opt}$.

Assume that player i moved to as an initial best-response $h_i^1$ to some $h^{opt}$. Then player j's best-response $h_j^1$ is such that $h_{opt}$ is between $h_i^1$ and $h_j^1$. Since $h_j^1$ is a BR, we know the market share of player $j$ increases (weakly).

For player $i$, from Proposition 1, $\mu_i = \frac{1}{2}(a_i + \delta_{ij})$.

$a_i$ remains the same, but $\delta_{ij}$ increase because $h_j^1$ went further away to the other side of $h_{opt}$, and as explained in the proof of Proposition 3, $\delta_{h_i^1, opt} \subset \delta_{h_i^1, h_j^1}$.

Therefore $\mu_i$ increases as well.

□

**Proposition 5 (Market share increases during competition).**

*Proof.* Assume the players started from $h^0$:

$\mu(h^0|h^0) = \frac{1}{2}a^0$

Let $(h_1, h_2)$ be anyt equilibrium.

$\mu(h_1|h_2) = \frac{1}{2}(a_1 + \delta_{12})$

We will prove $a_1 + \delta_{12} \geq a^0$:

Assume $a_1 + \delta_{12} < a^0$.

Then $a^0 + \delta_{h^0,2} > a_1 + \delta_{12}$, contradiction to $h_1$ being a best-response to $h_2$. □

**Corollary 2 (Welfare increases during competition).** This is immediate from Proposition 5 since $SW = \sum_i \mu_i$.

**Proposition 6**

*Proof.* Let $h_1, h_2$ be any PNE.

Assume for the sake of contradiction that $\mu(h_1|h_2) > 2 \cdot \mu(h_2|h_1)$.

From Proposition 1 we receive that $\mu(h_2|h_1) = \frac{1}{2}(a_2 + \delta_{21})$.

We will show that $\mu(h_1|h_1) = \frac{1}{2}a_1 > \frac{1}{2}(a_2 + \delta_{21}) = \mu(h_2|h_1)$:

W know that $\mu(h_1|h_2) > 2 \cdot \mu(h_2|h_1) \Rightarrow a_1 + \delta_{12} > 2 \cdot (a_2 + \delta_{21})$.

We also know $\delta_{12} \leq a_1$, by definition of partial discrepancy.

Therefore $a_1 + a_1 \geq a_1 + \delta_{12} > 2 \cdot (a_2 + \delta_{21})$

And so: $a_1 > a_2 + \delta_{21} \Rightarrow \mu(h_1|h_1) > \mu(h_2|h_1)$, contradiction to $(h_1, h_2)$ being a PNE.

□

# B. Additional theoretical results

## B.1. Characterization of our problem setting as a congestion game.

In Section 3, we mentioned that our problem setting is proven to have a PNE, a result shown by (Ben-Porat & Tennenholtz, 2019) through the use of an exact potential function. Additionaly, (Monderer & Shapley, 1996) show that every potential game is isomorphic to some congestion game; this connection however is not always readily evident. We show here the exact reduction of our problem setting to a congestion game, and highlight that the cost function is negative, which may be counterintuitive to more classic settings of congestion games.

**Observation 3.** *Our problem setting is reduced to the congestion game* $(N, M, (H_i)_{i \in N}, (c_j)_{j \in M})$ *Where:*

- *N is the number of players*

- *M is the samples in the training set upon which the players want to gain market share*

- $H_i$ *is the hypothesis class available to player* $i$

- $c_j(k) = -\frac{1}{k}$ *is the cost function assigned to each sample, where* $k$ *is the number of companies accurate on consumer* $j$

*Proof.* Firstly, we will show that the game that is defined above is indeed a congestion game. We can observe this almost immediately, as the cost function (while negative) is monotone increasing with $n_j$, and the cost is per-sample (equal for each player).

Additionally, the hypothesis class $H_i$ has a one-to-one function $a : H \to \mathbb{P}(M)$ which is $a(h) = \{(x, y) \in M : h(x) = y\}$. Therefore, each strategy $h$ is equivalent to the strategy $a(h)$ and this is a subset of the facilities $M$.

From the game that is defined, we receive a potential function $\Phi$ such that $\forall i, \ \Delta \Phi = \Delta C_i$.

The potential function is: $\Phi(\vec{h}) = \sum_{j=1}^{m} \sum_{k=1}^{n_j} c_j(k)$

(For each sample, we take the sum of $c(1), \ldots, c(n_j)$, and since $c$ is monotone increasing, minimizing the potential means minimizing both players being accurate for the same classifier)

Now, we will show that our problem setting reduces to this game by showing the equivalence between maximizing the player utility in the problem setting and minimizing the player cost in the above congestion game, meaning, $\forall i, \ \Delta U_i = -\Delta C_i$.

Let $s_i^k$ be the number of samples that player $i$ is accurate on along with $k - 1$ other players.

We observe that

$$C_i(\vec{h}) = \sum_{j \in a_i(h_i)} c_j(n_j(\vec{h})) = \sum_{k=1}^{n} s_i^k \cdot c(k) = -\sum_{k=1}^{n} s_i^k \cdot \frac{1}{k} = -U_i(\vec{h})$$

(where the middle equality comes from rearranging the samples in bins of how many other players were accurate, and and then the cost is constant in that bin).

$\square$

# C. Experimental details

## C.1. Data details

All of the experiments on real data were studied on 3 datasets: `compas-arrest`, `compas-violent`, and `adult`. The `compas` datasets originated from studies of recidivism in the United States (Angwin et al., 2016), and are used to predict if a criminal will be rearrested for general crimes and violent crimes, respectively. The `adult` dataset is used to predict whether the an individual's income exceeds $50K.

**Preprocessing Details:**

- **Adult:** The adult dataset was imported in python through the `uciml` library. All of the categorical features were one-hot encoded, and numerical features remain unprocesssed. To enable a balanced learning task, SMOTE resampling was

applied from the `imblearn` package to attain a 50% positive class ratio. After the above preprocessing, 10,000 samples were chosen randomly, resulting in a dataset with $n = 10,000$ samples and $d = 100$ features.

- **COMPAS-Arrest/Violent:** The COMPAS-Arrest dataset was preprocessed for analysis by Marx et al. (2020), and a copy of their csv files are included in their code. The csv files can be found at :
  https://github.com/charliemarx/pmtools/tree/master/data.
  Both datasets contain $d = 21$ preprocessed binary (previously one-hot encoded) features. The COMPAS-Arrest dataset contains $n = 6,172$ samples and has a positive class ratio of $45.5\%$.
  The COMPAS-Violent dataset also originally had $6,172$ samples, however the positive ratio was $88.8\%$. Therefore SMOTE upsampling was applied to the negative class to bring the positive ratio to 50%. The total number of samples for which we use COMPAS-Violent is then $n = 10,960$.

### C.2. Model Class details

For our empirical anlysis, we analyzed results of the experiments with 3 model class variants that were used as the effective strategy space of the service providers. We note that since the objective of this work is to understand the ability of providers to learn based on the importance of the *samples*, we kept the hyperparameter tuning minimal, so as not to forcefully overfit the data.

1. **Linear SVM:**

   - Hyperparameters: The regularization parameter $C = 1.0$. Other hyperparameters were left as default.
   - Hyperparameter tuning was performed on the values of $C$, but we observed no significant difference in the ability of providers to best-respond.
   - The model was implemented using the `LinearSVC` class from the `sklearn` package.
   - sample weights from our method were passed using the *sample weight* parameter of the *fit* method.

2. **XGboost:**

   - Hyperparameters:
     - Learning rate: $0.3$
     - Max tree depth: $6$
     - all other hyperparameters remained the default, in particular performing row and column subsampling of 1.
   - the loss metric used for boosting is *log-loss*
   - the model was implemented using the `XGBClassifier` class from the `xgboost` package
   - We note that some basic hyperparameter tuning was performed using a grid search, but default values yielded satisfactory results.

3. **Random Forest:**

   - Hyperparameters:
     - Number of estimators: $10$
     - Max tree depth: the default, meaning all nodes were expanded until all of the leaves are pure or contain a single sample.
     - all other hyperparameters remained the default.
   - the loss metric used for boosting is *log-loss*
   - the model was implemented using the `RandomForestClassifier` class from the `sklearn` package
   - Hyperparameters were minimally tuned, and the default values were primarily used.

### C.3. General implementation details

**Test and validation set.** For all experiments, the dataset was split into training, validation, and test sets. The test set comprised 20% of the data and was held out for final performance evaluation. The validation set, also comprising 20% of the data, was used for hyperparameter tuning when applicable. In cases where no hyperparameter tuning was performed, the validation set was not utilized, and so only the training and test sets were used.

**Experiment Splits.**   To ensure integrity and mitigate the effect of random variations in the data, each experiment was conducted over 10 random splits of the dataset. For each split, the data was shuffled and divided into training, validation, and test sets according to the above proportions. The reported results in the following sections include standard errors calculated across these 10 splits, providing an estimate of variability in the model performance.

**Code.**   All of our code is implemented in Python. All of our experiments are reproducible and attached as supplementary material.

**Hardware.**   All experiments were run in the PyCharm IDE on a single Macbook Pro laptop, with 16GB of RAM, and M2 processor, and with no GPU support. However, the experiments to create the table metrics were cumbersome on the IDE, and so the PyCharm heap size was raised to 8K MegaBytes in order to enlarge the stack. The total runtime for all the results takes roughly 12 minutes.

# D. Additional experimental results

## D.1. Main results for additional settings

In this appendix we showcase additional insights from our main results when tested on additional model classes.

### Table 3: XGboost performance

| | | Adult | | | | COMPAS-arrest | | | | COMPAS-violent | | |
|---|---|---|---|---|---|---|---|---|---|---|---|---|
| | | $\min \mu$ | $\max \mu$ | HHI | welfare | $\min \mu$ | $\max \mu$ | HHI | welfare | $\min \mu$ | $\max \mu$ | HHI | welfare |
| # providers | 2 | +1.1% | +2.1% | +3.2% | +1.6% | +20.5% | +39.0% | +69.3% | +29.7% | +26.7% | +54.6% | +100.0% | +40.7% |
| | 3 | +1.7% | +3.3% | +5.2% | +2.6% | +34.5% | +57.8% | +105.8% | +43.0% | +35.7% | +68.9% | +120.5% | +47.7% |
| | 4 | +2.4% | +3.8% | +6.2% | +3.1% | +35.3% | +66.3% | +116.1% | +46.4% | +36.3% | +66.1% | +125.8% | +49.8% |
| | 5 | +2.4% | +4.6% | +7.2% | +3.5% | +37.2% | +78.0% | +122.0% | +48.1% | +43.3% | +68.6% | +128.6% | +50.8% |
| | 6 | +2.0% | +5.0% | +7.2% | +3.5% | +40.2% | +73.1% | +124.0% | +49.1% | +43.4% | +69.3% | +130.8% | +51.6% |

**XGboost**   Table 3 shows the learning performance of the service providers when using XGBoost as the model class for training and inference. The details of the Xgboost implementation and hyperparameters can be found in Appendix C.2. A comparison of interest is the general improvements of the players relative to the Linear model class. We can notice that the welfare improvement, which is equal to the total market share of all players, is significantly lower for both the Adult and COMPAS-ARREST dataset. This does not indicate that the total welfare is lower, but rather that due to the increased expressiveness of the XGboost model, the starting welfare began at a higher value.

Another point of interest is how the HHI (the measure of imbalance in market share between the providers) persists across model classes, regardless of expressivity. This further highlights the importance of taking into account the market dynamics and the order of play when competing with other providers.

### Table 4: Random forest performance

| | | Adult | | | | COMPAS-arrest | | | | COMPAS-violent | | |
|---|---|---|---|---|---|---|---|---|---|---|---|---|
| | | $\min \mu$ | $\max \mu$ | HHI | welfare | $\min \mu$ | $\max \mu$ | HHI | welfare | $\min \mu$ | $\max \mu$ | HHI | welfare |
| # providers | 2 | +2.9% | +3.9% | +6.9% | +3.4% | +20.6% | +36.1% | +65.3% | +28.3% | +25.7% | +53.0% | +96.1% | +39.3% |
| | 3 | +4.1% | +5.8% | +10.1% | +4.9% | +30.9% | +58.7% | +98.9% | +40.3% | +34.3% | +71.8% | +118.9% | +46.9% |
| | 4 | +4.5% | +6.5% | +11.3% | +5.5% | +33.0% | +70.5% | +111.8% | +44.7% | +33.5% | +70.6% | +124.5% | +49.2% |
| | 5 | +4.5% | +7.5% | +12.5% | +6.1% | +34.2% | +81.0% | +117.0% | +46.0% | +41.7% | +75.1% | +128.3% | +50.5% |
| | 6 | +5.2% | +8.0% | +13.5% | +6.5% | +37.4% | +78.2% | +119.5% | +47.3% | +38.6% | +75.7% | +130.3% | +51.2% |

**Random Forest**   In a similar manner, Table 4 presents the results on competition where the providers are employing a Random Forest model, whose details can be found in Appendix C.2. We can observe that the results resemble those of the XGboost model class, which can be expected due to the similarity in nature of all Decision Tree models.

**Standard errors of experiments**   As mentioned in Appendix C, each experiment was run over 10 train-test splits and the metric values were averaged out over those splits. Table 5 portrays, for each metric, the maximum variation of the standard error among the three model classes analyzed. We note that all of the errors are below 5%, and most of the errors are well below 3%.

Table 5: Max standard errors of experiments across all model classes

| | | Adult | | | | COMPAS-arrest | | | | COMPAS-violent | | | |
|---|---|---|---|---|---|---|---|---|---|---|---|---|---|
| | | $\min \mu$ | $\max \mu$ | HHI | welfare | $\min \mu$ | $\max \mu$ | HHI | welfare | $\min \mu$ | $\max \mu$ | HHI | welfare |
| # providers | 2 | ±1.9% | ±4.4% | ±2.1% | ±2.6% | ±1.0% | ±2.6% | ±1.2% | ±1.3% | ±1.0% | ±2.9% | ±1.4% | ±1.1% |
| | 3 | ±0.8% | ±1.9% | ±1.0% | ±2.0% | ±1.2% | ±3.5% | ±0.8% | ±2.7% | ±0.8% | ±2.5% | ±0.8% | ±2.3% |
| | 4 | ±0.7% | ±2.0% | ±2.1% | ±4.2% | ±0.9% | ±2.9% | ±0.7% | ±3.2% | ±0.7% | ±2.2% | ±1.2% | ±3.2% |
| | 5 | ±0.9% | ±2.2% | ±1.1% | ±3.2% | ±1.0% | ±3.5% | ±0.7% | ±4.4% | ±0.7% | ±2.2% | ±0.8% | ±3.6% |
| | 6 | ±0.8% | ±1.9% | ±2.4% | ±3.4% | ±1.0% | ±3.2% | ±0.8% | ±4.4% | ±0.6% | ±2.1% | ±1.0% | ±2.7% |

## D.2. Competition Dynamics

**Market Share**   In order to provide a more comprehensive analysis of how the market shares of the providers move through the best responses in the competition, Figure 5 shows, for each number of players competing for market share, the shifts in market share of each provider based on their positions in the game (order of play).

We can observe a few key points:

1. **Market Stability.** Across all variations of the number of players, the market shares of the player converge almost immediately to their final respective values. This convergence occurs even before every player offered a single best-response, i.e, by the end of Round 1.

2. **Order of play.** In line with what is expressed in Section 6.2, The order of play when offering a best-response is important, and varies as a function of the number of players. For $n = 2$ providers, playing second can offer a significant competitive advantage. When shifting to markets with $n \geq 3$ providers, however, we observe a significant advantage to the 1st mover, as discussed in Section 6.2.

3. **General market share trend.** Regardless of the order of play, and as alluded to in Table 3, the market share rises for all players, which supports the empirical claim that providers that are competing are in a way *collaborating* to figure out how optimally divide the market.

4. **Generalization to an unseen test set.** The performance on the unseen test set closely mirrors that observed during training, highlighting the robustness of the competition method. The similarity in performance demonstrates that the model can calculate a best-response without overfitting, and underscores the ability of the framework to maintain accuracy in line with that expected from traditional ML methods.

## D.3. Welfare

**Welfare behaviors in additional settings.**   Figure 6 shows the social welfares across the model classes described in Appendix C.2, namely `LinearSVC, XGBoost,` and `RandomForest`. The test welfares are shown for the COMPAS-arrest dataset, and are portrayed for each model class and each game varying the nubmer of players. We can see that, as in Section 6.2, the welfare gets mazimized very early for the `Linear` model class, but hits a non-maximal plateau for the Decision Trees. This is another example of the non-monotonic nature of social welfare: in many cases providers having less expressiveness in their models will in fact benefit the consumers.

**Welfare across asymmetry in data.**   The social welfare trends across different levels of data representations can also behave in a surprising manner, as shown in Section 6.2 and explained in great detail by (Jagadeesan et al., 2024). As a portrayal of this phenomenon across different competition settings, Figure 7 plots the social welfares across experiments of 2,3, and 4 players, respectively. We can observe, that while no as blatant as with $n = 2$ providers, the general trend across all player formats is that the social welfare increases as an inverse proportion to the richness of the data representations.

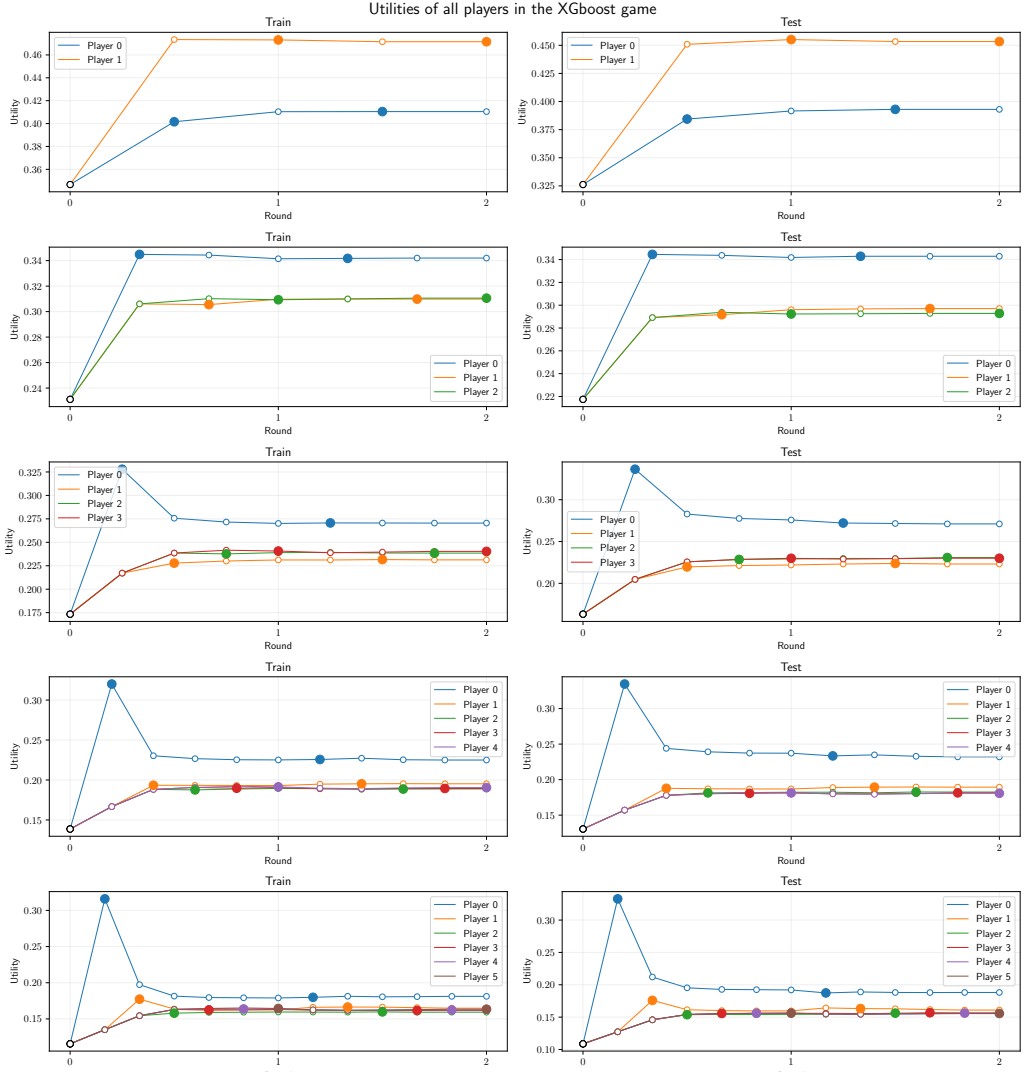

Figure 5: Competition Dynamics with XGboost models on the COMPAS-Arrest dataset. Utilities of individual players are shown for Train (Left) and Test (Right), in markets involving 2,3,4,5,and 6 players (Top to Bottom)

.

## D.4. Synthetic Data

In Section 6.1, we showed an example of the chicken dynamic between asymmetrical class-conditioned gaussians. Additionally, we performed an overlap analysis between symmetric gaussians, where for each measure of distance between the means, or overlap, we calculated the best-response thresholds and performance metrics. Figure 8 (Left) shows the above analysis on asymmetric gaussians, namely $\sigma_{-1} = 2, \sigma_{+1} = 1$. As in Section 6.1, at each overlap the thresholds of the players are initialized at $h_1^0 = h_2^0 = h_{opt}$, where $h_{opt}$ is the naively optimal classifier that maximizes accuracy on the distributions. Here too, and as is guaranteed by Theorem 3, the best-response dynamics converge after just one round. In this case of asymmetry between the class-conditioned gaussians, we notice a distinctly different behavior. while $h_1$ remains in the same proximity to $h_{opt}$ as in the symmetric case, threshold $h_2$ has a "tipping point", where it suddenly jumps to the far end of both distributions. From the accuracy graph we can also see a sudden drop in accuracy coming fromm model $h_2$. In regards to the market share, however, the provider that gets a spike in market share is in fact the one who played $h_1$ and remains close to the optimal, while the market share of $h_2$ simply shows a gradual increase, with no reference to a tipping point.

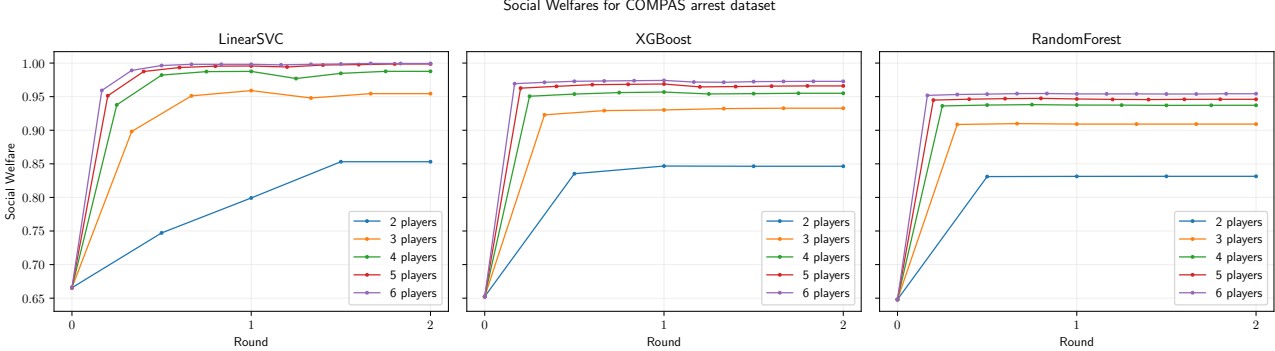

Figure 6: Social Welfares across model classes on the test set for the COMPAS arrest dataset. Each plot calculates the social welfare at each timestep and for each experiment that varies the number of players.

This phenomenon is understandable when we look at Theorem 3, that states that the best-response may be at the far end of the distributions. This occurs when either of the values $g^{-1}(1/2), g^{-1}(2)$ stops existing, where $g(x) = \frac{f_1(x)}{f_0(x)}$ is the MLR function. In these cases, the left or right threshold (depending on which value of $g^{-1}$ disappears) will continue to gain by moving to the far end of the interval. This is precisely what is shown in Figure 8 (Right); at a certain overlap ($\sim -1$), there is no threshold $h$ for which $g(h) = 2$, and so the gain in discrepancy for $h_2$ will continue to outweigh the loss in accuracy, and $h_2$ ends up at the far right end. The market share of $h_1$ is then suddenly increased, since while its accuracy remains the same, the discrepancy from $h_2$ is now significantly greater.

## D.5. Asymmetrical power between players

Another interesting question we can ask from the perspective of the learners/providers is, if a provider were to invest cost and effort to gain better data, how would this help them in competition? In naïve settings, i.e. single-provider markets, the answer to this is straightforward: more data = better accuracy. In accuracy markets, however, the specialization needs to be considered as well, and as we have seen (Section 6.2), the behavior of the market when altering the data quality can be counter-intuitive. To measure the gain to be had when improving data, we ran experiments where one of the providers has access to more features than its counterpart/s. For each of 2 possible move positions (1st or 2nd), two metrics were considered:

1) The provider's gain in market share over himself if he weren't to improve his data (i.e. the data is identical for all parties), which measures the provider's marginal gain from improving data

2) The provider's gain in $\Delta\mu$ from the next-best provider versus the setting where he didn't improve his data, which measures

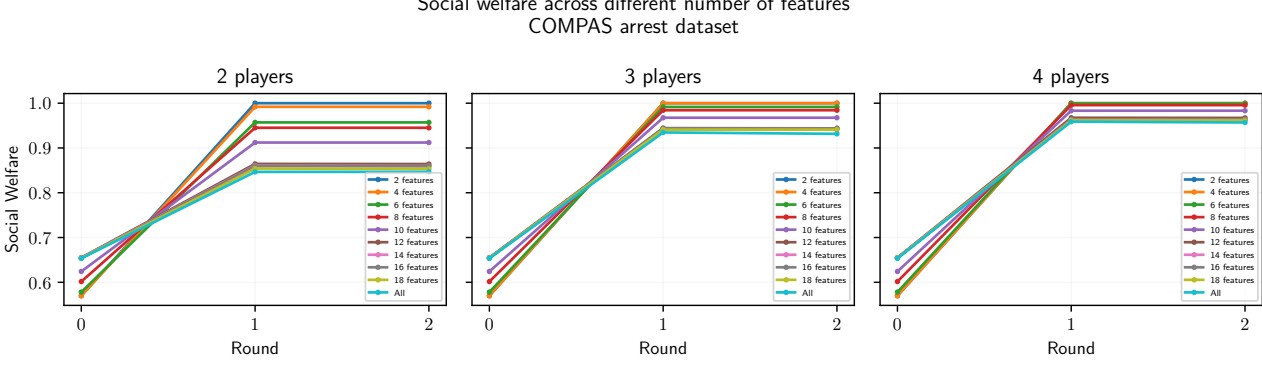

Figure 7: Comparison of social welfares on the COMPAS arrest test set when varying the number of features available to use for model training (and inference).

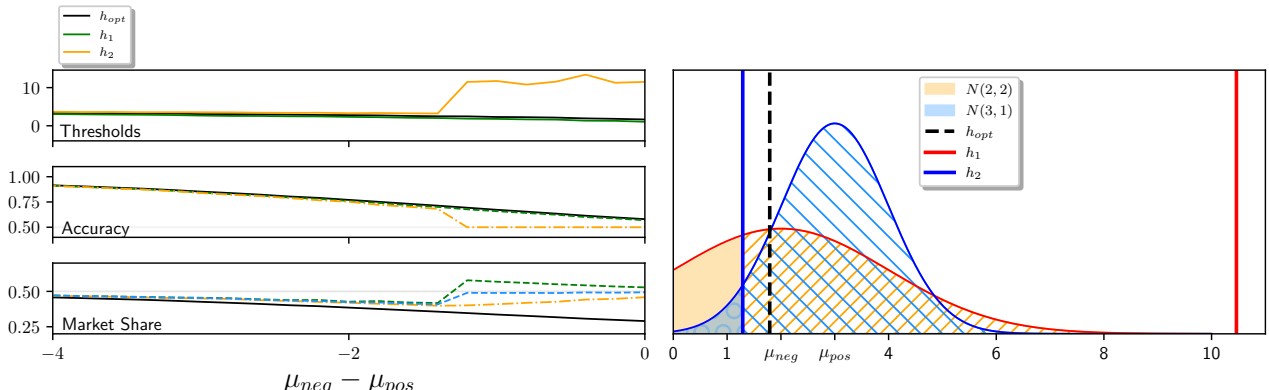

Figure 8: Dynamics of the best-response game on threshold classsifiers with asymmetric gaussians. The metrics at each level of proximity between the gaussians are shown (Left). The thresholds at the "tipping point" (Right) can be seen to be very far apart. The hatches represent the sectors of exclusivity.

the impact of the data investment on the concentration of the market as a whole.

Table 6 shows the above metrics for markets with 2,3, and 4 providers, and for various possibilities of data improvement.

We can observe a few interesting trends:

1. $\Delta\mu$ **from regular setting.** One of our central insights from Section 6.2 is that when $n = 2$ providers, moving second is beneficial, and when $n > 2$, the opposite is true, and moving first is better. In the case of investing in better data, and when comparing the provider's market gain vs. himself in a regular setting, we see an inverse effect.

   For $n = 2$, better data creates market gains only when you are the first mover, as in certain lopsided markets the gain is $> 12\%$. For example, in the case where the better-data provider has 15 features, and the other providers have 3 features, we can observe a 12% gain, which measures the benefit of investing in the additional 12 features.

   When the provider moves second, however, investing in more data does not translate to higher market share, in fact the provider loses significant market share, and would have been better off retaining the same primitive data as the other competitors. This phenomenom gets exacerbated further the more the provider invests in better data; For example, if one were to utilize all 21 features of the dataset when the competitors have access to only 9 features, the advatnaged provider would see a -12.35% loss in market share.

   For markets where $n > 2$, it is the other way around. When the advantaged provider moves first, he may see a decrease in market share from the regular setting where he didn't gain extra data; When moving 2nd, the data gain proves helpful. This stands in polar contrast to the case where $n = 2$, and perhaps understandably so: It seems that wherever the providers have an initial advantage when the data is symmetrical, they would lose that advantage when investing in more data, perhaps hinting at the idea of decreasing marginal returns in investments.

2. $\Delta\mu$ **from the next-best provider.** When comparing the difference across data-variation experiments in market shares between providers, we notice that the trend behaves similarly to how we have seen in the order-of-play results of Section 6.2. We can observe that, interestingly, if a provider improved his absolute market-share relative to himself, this doesn't translate to the provider improving his market share relative to others. Take for example the cases where $n > 2$ and the provider moves 2nd. As stated above and as can be seen in Table 6, the added data advantage in this setting helps the provider gain in absolute market share. The market gain relative to the other providers, however, has an inverse result, and in many cases the competitors end up with a better overall utility. This tells us that in these settings, when the advantaged provider goes second, the total welfare (as the sum of individual market shares) increases.

Table 6: Asymmetrical power between players. Results are shown for markets with 2,3,and 4 providers, respectively. Rows are shown for each choice of number of features for the provider with better data,and the columns for the number of features of the other providers. Table results are shown for using XGboost trees on the `compas-arrest` dataset, that contains 21 features in total.

**# features of the worse data**

| # providers | move position | # features: better data | $\Delta\mu$ from regular setting | | | | | | $\Delta\mu$ from next best provider | | | | | |
|---|---|---|---|---|---|---|---|---|---|---|---|---|---|---|
| | | | 3 | 6 | 9 | 12 | 15 | 18 | 3 | 6 | 9 | 12 | 15 | 18 |
| 2 | first | 6 | 3.20% | | | | | | -14.37% | | | | | |
| | | 9 | 8.96% | 5.34% | | | | | -1.17% | -3.97% | | | | |
| | | 12 | 12.82% | 9.41% | 4.76% | | | | 14.04% | 12.21% | 1.01% | | | |
| | | 15 | 12.66% | 9.15% | 4.99% | 0.48% | | | 14.48% | 12.41% | 2.17% | -10.96% | | |
| | | 18 | 11.85% | 8.76% | 5.43% | 0.60% | -0.33% | | 13.46% | 11.86% | 3.69% | -9.43% | -10.63% | |
| | | 21 | 10.10% | 7.01% | 2.53% | -3.44% | -3.42% | -3.11% | 9.98% | 8.02% | -1.16% | -14.89% | -14.12% | -13.96% |
| | second | 6 | -4.98% | | | | | | 20.32% | | | | | |
| | | 9 | -8.78% | -4.38% | | | | | 17.55% | 14.03% | | | | |
| | | 12 | -13.36% | -5.26% | -8.40% | | | | 15.92% | 17.19% | 5.15% | | | |
| | | 15 | -13.95% | -5.62% | -9.91% | -0.33% | | | 15.91% | 17.74% | 3.98% | 12.64% | | |
| | | 18 | -14.55% | -6.17% | -10.28% | -1.32% | -0.81% | | 14.88% | 17.09% | 3.53% | 11.06% | 10.35% | |
| | | 21 | -16.46% | -8.38% | -12.35% | -3.29% | -2.97% | -2.27% | 9.92% | 13.09% | -0.89% | 5.76% | 5.74% | 7.20% |
| 3 | first | 6 | -0.48% | | | | | | 51.86% | | | | | |
| | | 9 | 0.25% | 1.93% | | | | | 57.14% | 55.02% | | | | |
| | | 12 | -6.79% | -2.88% | -1.58% | | | | 40.07% | 46.39% | 30.02% | | | |
| | | 15 | -4.65% | -3.75% | -1.33% | -0.41% | | | 45.18% | 44.24% | 31.88% | 18.26% | | |
| | | 18 | -3.09% | -2.55% | -1.53% | -0.99% | -0.65% | | 51.18% | 47.61% | 31.82% | 17.21% | 18.85% | |
| | | 21 | -10.45% | -7.13% | -3.11% | -2.65% | -2.56% | -1.44% | 30.02% | 37.01% | 30.14% | 16.38% | 17.40% | 17.42% |
| | second | 6 | 1.43% | | | | | | -27.44% | | | | | |
| | | 9 | 9.68% | 9.78% | | | | | -11.53% | -8.80% | | | | |
| | | 12 | 15.39% | 12.55% | 5.81% | | | | -7.08% | -6.05% | -7.29% | | | |
| | | 15 | 15.10% | 12.33% | 5.51% | 0.47% | | | -8.34% | -7.49% | -8.85% | -15.49% | | |
| | | 18 | 14.97% | 12.24% | 5.40% | 0.84% | -0.25% | | -8.39% | -7.48% | -9.09% | -14.97% | -16.66% | |
| | | 21 | 14.63% | 11.25% | 5.74% | -0.04% | -0.11% | -0.42% | -7.84% | -7.53% | -8.84% | -13.25% | -13.49% | -12.83% |
| 4 | first | 6 | 1.09% | | | | | | 24.08% | | | | | |
| | | 9 | -12.59% | 5.66% | | | | | -8.55% | 13.93% | | | | |
| | | 12 | -8.11% | 3.04% | 0.82% | | | | -1.02% | -0.21% | 16.63% | | | |
| | | 15 | -7.90% | 2.01% | -3.05% | -0.67% | | | -0.51% | -1.72% | 9.94% | 16.75% | | |
| | | 18 | -8.18% | 2.04% | -2.28% | -0.90% | -0.60% | | -1.00% | -1.90% | 11.50% | 16.96% | 15.68% | |
| | | 21 | -7.66% | 1.64% | -3.96% | -2.22% | -2.36% | -1.32% | -0.35% | -2.98% | 8.00% | 12.08% | 10.02% | 12.10% |
| | second | 6 | 6.64% | | | | | | -11.54% | | | | | |
| | | 9 | 14.80% | 11.58% | | | | | -11.82% | -0.20% | | | | |
| | | 12 | 21.42% | 16.05% | 9.82% | | | | -4.49% | -3.83% | -9.99% | | | |
| | | 15 | 21.13% | 15.91% | 9.61% | 0.00% | | | -4.75% | -3.78% | -10.45% | -16.16% | | |
| | | 18 | 21.03% | 15.72% | 9.24% | 0.10% | 0.46% | | -4.97% | -4.63% | -10.51% | -16.32% | -15.31% | |
| | | 21 | 19.48% | 13.98% | 8.23% | -0.69% | -0.73% | -0.35% | -7.57% | -6.54% | -13.06% | -18.51% | -18.15% | -17.31% |

