# OpenReview forum: "A Market for Accuracy: Classification Under Competition"
_ICML.cc/2025/Conference — ICML 2025 poster_

### Official Review · Reviewer_GVZQ · 2025-03-11

**Overall Recommendation:** 3

**Summary:**

This paper examines a machine learning (ML) model market for classification tasks. Specifically, the authors assume that multiple ML model providers compete for market share. Each user in the market randomly selects an ML provider that delivers an accurate prediction, and the market share of each provider is defined as the expected number of users choosing its model. The authors analyze the fundamental properties of best-response classification in simplified 2 × 2 accuracy markets and markets with threshold classifiers. They show that best-response classification can benefit both ML providers and consumers. Additionally, they propose an approach to effectively solve best-response classification using real finite samples and predictions from other ML providers. Synthetic and real-world experiments validate the theoretical findings.

---

I remain the score unchanged after rebuttal.

**Claims And Evidence:**

**(Pro)** The theoretical results are generally sound and are supported by both theoretical analysis and experiments.

**(Con)** Best-response classification may be impractical in real-world settings since ML providers typically lack access to competitors’ predictions. While Equation (10) only requires information on the number of accurate predictions for each sample, obtaining this data in practice remains challenging. It would be helpful for the authors to discuss the implications of cases where this information is unavailable or biased.

**(Con)** The theoretical results are primarily based on the $N=2$ setting. However, in real markets, the number of ML model providers is often much larger. Extending the theoretical analysis to a general $N$-provider setting would strengthen the paper.

**Essential References Not Discussed:**

No essential references appear to be missing.

**Experimental Designs Or Analyses:**

**(Pro)** The experiments are well-designed to validate both the theoretical findings and the proposed method. They also explore more general settings beyond those covered in theory, which is valuable.

**Methods And Evaluation Criteria:**

**(Pro)** The proposed method is theoretically sound and validated through experiments.

**Other Comments Or Suggestions:**

**(Pro)** The paper is generally well-written.

**Other Strengths And Weaknesses:**

No additional strengths or weaknesses were identified.

**Questions For Authors:**

See the concerns listed above.

**Relation To Broader Scientific Literature:**

Compared to Ben-Porat & Tennenholtz (2019), this work focuses specifically on classification problems and provides a more detailed analysis of best-response dynamics.

**Theoretical Claims:**

I did not verify the details of the theoretical claims, but the results appear to be generally sound.

---

> ### Author Rebuttal · Authors · 2025-03-31
>
> Thank you for the insightful feedback! We are pleased that you appreciated the soundness of our results and found the empirical results to complement them nicely. We would like to address the questions laid forth in your review, and hopefully alleviate some concerns:
>
> **”Best-response classification may be impractical in real-world settings since ML providers typically lack access to competitors’ predictions”**
> Indeed, our analysis relies on the simplifying assumption that players have complete knowledge of $\\kappa_{-i}$, the number of other correct firms per $x$. This, however, need not be taken to imply that players share such information. Instead, we view this as simplifying a process in which each player has access to (some) information regarding competition. For example, there are markets where firms are likely to have a good sense of which user groups the other firms excel on or target. Another example are cases where a firm performs market research and obtains exact information on a subset of users. A third example is firms having coarse information, such as knowing whether users subscribed to a different provider.
> To investigate this idea, **we have added an additional experiment on partial and coarse information**, in which firms have inexact estimates $\hat{\kappa}_{-i}$, and therefore learn with inexact weights. We consider two settings:
> - **Coarse:** For each user, a player has knowledge only of whether there is at least one other accurate player, i.e., $\\hat{\\kappa}_{-i}$ is either 0 or rounded down to 1.
> - **Partial:** Players know the true $\kappa_i$ for a random subset of users, and use this to make inference about the remaining $\\hat{\\kappa}_j$ (we use kNN).
> Our results suggest that despite partial or coarse information, overall trends are preserved. However, the cost of misinformation is that social welfare is lower (and hence also total market share). For *coarse*, this reduction is 37% for $n=3$, and up to 45% for higher $n$. For *partial*, we see for $n>3$ that market research on just 100 users results in a welfare gain that is >70% of that under full information, suggesting that extrapolation can work well in our setting.
> Additionally, this gain increases with the size of user information. We found that providers are able to learn near-optimal best responses even when only having done market research on 20% of the population. The gain in welfare is summarized in the following table for the COMPAS-arrest dataset:
>
> |   |  # research points| 100  (2.7%) | 200  (5.4%) | 400 (10.8%) | 800 (21%) | Full information |
> |:-----------:| --|:----------:|:----------:|:----------:|:----------:|:----------:|
> | **# providers**   |   |  |  |  |   |    |
> | $n=2$    |   | +17% | +21%   | +23%  | +26%   | +29%   |
> | $n=3$    |   | +25%   | +33%  | +35%  | +37%   | +42%  |
> |   $n=4$  |   | +34%  |+38%  | +42%  | +44%  | +46%  |
> | $n=5$    |   | +38%  | +42%  | +45% | +47% | +48%  |
> | $n=6$    |   | +41%  | +44%  | +46%  | +48%  | +48%  |
>
> These findings show promise to the approach of extrapolating to the uninformative user sectors, and suggest that even in real-world settings our approach can be worthwhile.
> We will be sure to present these new results using the extra page of the final version, and provide full details of the experimental setup and complete results in the Appendix.
>
> **”Extending the theoretical analysis to a general N-provider setting would strengthen the paper”**
> Further analysis for multiple players is certainly interesting, and it is indeed our hope that this work inspires future work in multi-player settings. We will note though that this is not without challenges. Take, for example, our notion of partial discrepancy ($\\delta$), which offers a simple expression for utility in 2-player settings. As $n$ grows, the number of partial discrepancies between any 2 sets of players grows exponentially, and so extending our results to general $n$ becomes complex. We would also like to point out that the works of Ben-Porat & Tennenholz (upon which we base our setting) began with a work on 2 players (2017), and later released an entirely separate work to discuss the $n>2$ player setting (2019).
>
> Thanks again for the helpful feedback. We are happy to discuss further any follow-ups regarding the new experiments and topics discussed above. If you found these to your satisfaction, we would greatly appreciate it if you would consider raising your score.

---

> > ### Comment · Reviewer_GVZQ · 2025-04-03
> >
> > Thank you for the response. I will keep my score unchanged.

---

> > > ### Author Response · Authors · 2025-04-07
> > >
> > > Thank you for your positive review and enlightening feedback!

---

### Official Review · Reviewer_jrJ7 · 2025-03-13

**Overall Recommendation:** 3

**Summary:**

This paper defines a simplified economic model to consider how the dynamics of an “accuracy market” between multiple firms would play out. In this model, each firm has a model that makes predictions about a user, and users are assumed to choose a provider with the highest accuracy. The paper lays out several theoretical and empirical results. First, it shows that firms must consider both their own accuracy and the accuracy of other players in the market, adopting a weighted objective approach. Second, it shows that adopting this strategy benefits both the model providers and the users, as defined through a welfare metric. It then considers several datasets and experiments, including both with synthetic data and simple real world datasets.

**Claims And Evidence:**

I believe the claims made in the paper regarding the behavior of these idealized markets are supported both by the proofs and empirical experiments. My primary reservations with the work are regarding the assumptions and relevance to ICML, as discussed in the “Other Strengths and Weaknesses” section below.

**Essential References Not Discussed:**

I am not aware of any essential references that were not discussed.

**Experimental Designs Or Analyses:**

My overall impression of the experimental designs is that they are sound, but I did not have the opportunity to dive deeply into any code or data to verify further.

**Methods And Evaluation Criteria:**

As far as I can tell, the experimental setups make sense for the idealized market setting proposed in the paper.

**Other Comments Or Suggestions:**

N/A

**Other Strengths And Weaknesses:**

The strengths of this paper are its clarity and novelty. The paper does a good job of showing why, in this idealized setup, firms need to consider their competitors. Based on my limited search, it also seems as though this is the first paper to consider a market-based approach in the classification regime (although they do cite work that does similar treatment for regression models). Several results are formally proven, and I do think it is likely to inspire discussion among the right set of readers.

The primary weaknesses of this work are its relevance to ICML and its practicality. First, the work seems like it would be more appropriate for an economics venue than a machine learning one. While it does introduce an economic formalism to a machine learning topic, it feels much heavier on the economics side. I acknowledge that this may be due to my own lack of knowledge regarding economic approaches to ML, but I had a hard time following several parts of the paper despite being an experienced ML practitioner. I also note that the primary work that the paper builds on, Ben-Porat and Tennenholtz, was published at the ACM Conference on Economics and Computation.

Second, the setting seems quite idealized for how ML actually happens in practice. For example, I noted right away in equation (1) that the choice to have a model making one prediction per user seems odd - if there are firms competing over users (for example, in social media), they would likely be making several predictions about a user’s preferences, and those predictions in aggregate would influence the user’s choice of firm. There also does not seem to be much consideration of the effect that number of users has on the accuracy itself. Accuracy is not a quantity that can be purchased as in a traditional market - instead, the main way to get more accuracy is to have more users and thus get more data. I may have missed it, but I do not think that the model accounts for this relationship. I suspect that if it were taken into account, the market timing results would be different - a first mover would be more likely to be able to improve their accuracy more quickly because they have more data than competitors.

Because of its simplifying assumptions and shaky relevance to ICML, I would recommend that this paper be submitted to a different venue.

**Questions For Authors:**

1. Why assume that there is only one prediction per user in equation 1 (and 7)? Are there any settings where this would be true?

2. Is it possible to summarize the results in table 2 in a more visual way? At the moment it is very hard to understand what is going on or easily spot the trends that are described in section 6.2.

**Relation To Broader Scientific Literature:**

The work seems to be primarily related to other work on algorithmic markets. In particular, the work seems to build on the 2019 Ben-Porat and Tennenholtz paper on regression.

**Theoretical Claims:**

I did not check the details of the proofs as they are outside my area of expertise.

---

> ### Author Rebuttal · Authors · 2025-03-31
>
> Thank you for your review and efforts. We have addressed your concerns in our response below. As for the issue of whether our paper is a good fit for ICML – which seems to be a main concern – we hope that our response, combined with the other reviews for our paper, help in establishing its relevance and appropriateness. Given this, **we are hopeful that you will be willing to reconsider your position regarding acceptance**, and will gladly answer any further questions you may have.
>
> **Relevance to ICML:**
> ICML includes a large sub-community of researchers that study questions at the interface of learning and economics, with `game theory` being one of the conference’s stated topics of interest (see this ICML’s call for papers). Many of the papers we cite and share connections with have been published in leading machine learning venues (see references). In fact, we view our work as lying *more on the machine learning side* than typical works in this space. As for Ben-Porat & Tennenholtz (2019), please note that it is in fact a follow-up to Ben-Porat & Tennenholtz (2017) – which was published in NeuIPS.
>
> **”The setting seems quite idealized for how ML actually happens in practice.”**
> Studying learning in economic settings requires making simplifying assumptions. This is especially true when aiming to establish a theoretical understanding – which is one of our goals. This also allows us to build on earlier results (e.g., that best-response dynamics reach equilibrium) that rely on similar assumptions. Despite these limitations, one of our key points is that naively optimizing accuracy can be a bad strategy. If anything, this portrays conventional learning frameworks as “idealized” when confronted with an economic reality. Our hope is that our work serves as a step towards further exploration of learning in practical market settings.
>
> **”Why assume that there is only one prediction per user..?”**
> While not applicable to all scenarios, this modeling choice is appropriate for many real-world settings where each user makes only one decision and hence is in search of a single prediction. For instance, in housing or car sales, a user typically sells a single house or car, and therefore looks for a single accurate prediction about the worth of the entity (for example from Zillow as a real estate valuation predictor). This leads to a  well-defined $(x, y)$ pair per user.
> Other examples are cases where users interact with multiple items (e.g., recommending multiple products in social media sites), but the outcome of interest often remains singular at the user level. For example, a user may receive multiple predictions $(x, z)$, but only one true outcome $y$ (e.g. user satisfaction). This aligns with our setting where the goal is to model a single, well-defined outcome per user.
> We agree that there are also settings in which users are associated with multiple labels. One idea is to model user decisions based on the probability of correct prediction $P(\\hat{y} = y)$ (e.g., quantal response). However, equilibrium results for such a model remains an open question and is beyond our current scope. We believe our framework provides a solid starting point and can be extended to incorporate such user dynamics in future research.
>
> **”The main way to get more accuracy is to have more users and thus get more data”**
> This is a good point. In our work sample size comes in indirectly through competition: for player $i$, if other players are correct on many points, then these will get small weights in $i$’s objective, and therefore its *effective* data size (see [1]) will be small. Our setup abstracts away the direct effect of competition on data size – this is an intentional design choice, intended to enable tractable analysis of our main phenomena. One justification is that accuracy gains from increased sample size typically exhibit diminishing returns; i.e., once a reasonable number of samples are obtained, adding more samples has only a mild effect on accuracy (see e.g. [2]). In other words, sample size effects are important mostly in the small-data regime, which is not our focus.
> A final note is that in our setup, the goal of learning is *not* maximizing accuracy, but rather, market share. Our results show that aiming for maximal accuracy can be a poor strategy, and that it is often better to focus efforts on a small subset of the population than to seek better accuracy on as many users as possible.
>
> **”Is it possible to summarize the results in table 2 in a more visual way?”**
> To complement Table 2, we have already illustrated the full best-response dynamics in Figure 5 and provided a detailed discussion of such in Appendix D.2. If you have specific suggestions for further improving clarity, we are happy to consider them and will gladly add them to the final version if applicable.
>
> [1] Pustejovsky (2019). Effective sample size aggregation.
> [2] Kaplan, Jared, et al.  (2020)  Scaling laws for neural language models.

---

> > ### Comment · Reviewer_jrJ7 · 2025-04-04
> >
> > Thank you for the rebuttal and the clarification on the suitability of this topic for ICML. In light of the response and the other reviewers' reviews, I have updated my score to a 3.

---

> > > ### Author Response · Authors · 2025-04-07
> > >
> > > Thank you for revisiting your score and for your thoughtful feedback on our work. We really appreciate it!

---

### Official Review · Reviewer_n7A1 · 2025-03-15

**Overall Recommendation:** 4

**Summary:**

This paper studies equilibria in marketplaces when multiple firms are competing with each other for consumers. As compared to the prior literature in this space, this paper focuses on the dynamics of learning an equilibrium between two firms, as well as the impact on consumers and markets. Interestingly, this paper shows that firms can converge to a stable equilibrium relatively quickly, shows that firms can implicitly coordinate which markets they enter, and that increased competition helps consumers. It concludes with experiments on synthetic and real data. The paper is pretty clearly written and addresses related literature well.

**Claims And Evidence:**

Yes - this paper is clearly written and makes a compelling contribution to the study of how firms can converge to (anti)-competitive behavior.

**Essential References Not Discussed:**

See above

**Experimental Designs Or Analyses:**

N/A

**Methods And Evaluation Criteria:**

Yes

**Other Comments Or Suggestions:**

N/A

**Other Strengths And Weaknesses:**

It would have been interesting to have deeper discussion of the effect of capacity on competition (discussed at the end of section 6).
Similarly, it also could have been helpful to have deeper discussion of how the main results of the paper would change given a wider model of classifiers (e.g. where MLR isn't satisfied).

**Questions For Authors:**

See above

**Relation To Broader Scientific Literature:**

One high-level takeaway from the paper is that firms effectively coordinate by independently deciding to compete in different markets. This occurs even though firms move sequentially and don’t coordinate explicitly. Given this result, it would be helpful if authors address how their work connects with work on algorithmic collusion, e.g. https://arxiv.org/pdf/2409.03956.

**Theoretical Claims:**

N/A

---

> ### Author Rebuttal · Authors · 2025-03-31
>
> Many thanks for your positive review! We are pleased that you found that the paper was clearly written and that you enjoyed our result showing how competition can induce anti-competitive behavior. In line with this, we would like to address some of your feedback below:
>
> **”It would be helpful if authors address how their work connects with work on algorithmic collusion”**
> Thank you for raising our attention to this work! It is interesting to see another example of cooperative behavior arising implicitly in a setting of competition. It seems that they show that under certain families of stateful online algorithms, multiple competitors can actually retain monopolistic prices, which would benefit all of the providers. While our focus here is how machine learning prediction can be optimized under competition, their positive result in the area of **pricing optimization** is certainly interesting to see, and gives thought to how we can incorporate pricing into future work. More specifically, it would be a neat extension to define a price parameter for each provider, and as such the user choice can be according to which provider gives the best price-to-accuracy tradeoff.
> With that said, there are some fundamental differences between their work and ours which bear consideration:
> - As mentioned above, our focus is entirely on how **machine learning predictors** can implicitly coordinate under competition (through the *weighted-accuracy* objective laid out in Section 5). The work on Algorithmic Collusion does not consider the effect that product relevance has on the user choices, and therefore there is no consideration for how predictors play a role in the strategies of the players.
> - In our setting we assume best-response dynamics, where the players respond optimally to the **current** strategies of the competitors. Indeed in the pricing model of their work, if players best-responded locally at each timestep, then the dynamics would converge to expected competitive behavior.
> - As stipulated in the performativity paragraph at the end of Section 5, our setting is **stateless** (at least for $n=2$), whereas the Algorithmic Collusion work is **stateful** in that it takes into consideration the pricing in previous timesteps. This may also explain the positive result under the family of no-regret algorithms, which are different from best-response dynamics.
>
>
> **“It would have been interesting to have deeper discussion of the effect of capacity on competition”**
> We appreciate your interest in how the capacity of data representation affects the overall welfare. It is indeed a counter-intuitive result, and one reasoned about at length in the work of Jagadeesan et al. (mentioned in the paper), albeit in a different (though related) setting. Our intention here was to show how the result of “lower data representation resulting in higher welfare” shows itself in our setting as well, as it seems to be a central characteristic of competition in machine learning. We would also like to note that the main paper shows this result for $n=2$ players, and results for $n=2,3,4$ players are shown in Appendix D.3 (Figure 7).
>
> **“How [would] the main results of the paper change given a wider model of classifiers?”**
> Thank you for this feedback. To make sure we fully understand your concern, is your intention regarding the theoretical results for $n=2$ players? Regarding our results on equilibrium, market share dynamics, and welfare, indeed when extending to general model classes and for multi-dimensional data (Section 4.3 in the paper), the convergence results may not hold. Our empirical results, however, show that in general model settings, convergence still occurs almost immediately, within two rounds. The market share and welfare results hold strongly as well, suggesting, as you may have alluded to here, that there is opportunity for future work to extend our results to more general settings, as we did in the 1-dimensional case.

---

### Official Review · Reviewer_37Jm · 2025-03-24

**Overall Recommendation:** 3

**Summary:**

The paper studies a market where multiple model providers compete to provide accurate predictions to as many users (points) as possible. The theoretical analysis reveals several interesting insights. The main insight is that naively maximizing for accuracy is not optimal for either player. For example, if there are two players and both use the optimal classifier that is accurate on say 80% of the points. Since they are accurate on the same points their payoffs are divided into half. They can increase their payoffs by deviating from the optimal classifier and being accurate on subsets of points (subpopulations) exclusively. This dynamics of the market improves the payoffs of the players and maximizes the welfare of users. Thus overall the market benefits all parties despite the competition. These insights are also illustrated through experiments on synthetic and real datasets.

**Claims And Evidence:**

The claims made in the submission are supported by clear and convincing evidence.

**Essential References Not Discussed:**

This is fine.

**Experimental Designs Or Analyses:**

Both synthetic and real data experiments are designed to evaluate the theoretical claims. Overall, there are no issues in the design of experiments, and they support the theory. One issue is the absence of error bars in the numerical results.

**Methods And Evaluation Criteria:**

The methods and evaluation criteria are reasonable.

**Other Comments Or Suggestions:**

1. Introduction is easier to follow after reading the setup and some of the empirical results. Otherwise, the setting and the dynamics of the game are not clear which makes it difficult to follow the introduction. I’d suggest making the introduction crisper and also introducing the setting and some examples (e.g. Figure 1) earlier.


2. In section 4, it would help to list the main claims clearly. It is likely that readers can get lost in a series of propositions and theorems.

**Other Strengths And Weaknesses:**

S1. The paper provides an interesting analysis of accuracy market dynamics. Naive intuition suggests each player’s best interest is in deploying the model with the highest accuracy. The analysis shows it is not true and all parties can maximize their utility by deploying models that are accurate on exclusive sets of points.

S2. The paper is well-written with a clear problem setup and analysis. I appreciate the analysis in a 2-player setting and with 1-d threshold-based classifiers. The synthetic experiment results in Figure 1 helps in getting to the core of the market dynamics and why the claims made in the paper make sense.


------


W1. The biggest weakness is the assumption that the players know the predictions of other players on all points and they are learning models on the same data points (eq. 11). In practice such information is private to the players and they may not share the same data points.

W2. While the analysis in 2-player setting is good as a first step and easier to understand. Analysis in multi-player settings would make the paper stronger.


W3. I like the paper for the insights but I am not sure about the practical relevance as the setting and assumptions seem too unrealistic. Grounding in a specific application could help here.

**Questions For Authors:**

1. In practice precise information of other players may not be known. In such cases how will the analysis and implementation unfold? Are there applications where the assumptions made in the paper are realistic?

2. Can the empirical results be provided for realistic applications?

**Relation To Broader Scientific Literature:**

The paper provides novel insights into the accuracy markets.

**Theoretical Claims:**

I checked a few (Proposition 1 and Theorem 1). I do not see any major issues in other claims.

---

> ### Author Rebuttal · Authors · 2025-03-31
>
> Thank you for your feedback! We are encouraged that you found the paper well-written and that you appreciate our analysis and experiments.
>
> **”The biggest weakness is the assumption that the players know the predictions of other players…”**
> Indeed, our analysis relies on the simplifying assumption that players know the number of other correct classifiers. This, however, need not be taken to imply that players share such information. Instead, we view this as simplifying a process in which each player has access to (some) information regarding competition. For example, there are markets where firms are likely to have a good sense of which user groups the other firms excel on or target. Another example are cases where a firm performs market research and obtains exact information on a subset of users. A third example is firms having coarse information, such as knowing whether users subscribed to a different provider.
> To investigate this idea, **we have added an additional experiment on partial and coarse information**, in which firms have inexact estimates $\\hat{\\kappa}_{-i}$, and therefore learn with inexact weights. We consider two settings:
>
> - **Coarse:** For each user, a player has knowledge only of whether there is at least one other accurate player, i.e., $\\hat{\\kappa}_{-i}$ is either 0 or rounded down to 1.
> - **Partial:** Players know the true $\kappa_i$ for a random subset of users, and use this to make inference about the remaining $\\hat{\\kappa}_j$ (we use kNN).
>
> Our results suggest that despite partial or coarse information, overall trends are preserved. However, the cost of misinformation is that social welfare is lower (and hence also total market share). For *coarse*, this reduction is 37% for $n=3$, and up to 45% for higher $n$. For *partial*, we see for $n>3$ that market research on just 100 users results in a welfare gain that is >70% of that under full information, suggesting that extrapolation can work well in our setting. Please see the table attached in our response to reviewer GVZQ for a more detailed comparison.
> We will add these new results to the Appendix.
>
> **”In practice … players may not share the same data points”**
> This is a fair point (and one we mentioned in our broader impact section). From a learning perspective, it is certainly possible to define a market where different firms have differing datasets, and in this setting our method would be valid and well-defined. However, from a game theoretic perspective, once exact datasets differ, then the market is no longer a congestion game, and its analysis becomes much more involved. We are familiar with only one work that aims to extend congestion games to settings where resources are not shared [1], but this requires making strong structural assumptions and the results are limited. Nonetheless, we agree that exploring a setting with differing datasets is of practical value and can be interesting as future work.
>
> **”Grounding in a specific application could help here”**
> Thank you for this recommendation. Consider a media platform, such as Netflix, whose value lies (in large part) in their ability to recommend relevant content to their users. Given their knowledge of the potential user base, they would like to create a recommendation system that maximizes subscriptions. Using their knowledge of the performance of the competitors (Disney Plus, etc.), whether through general, coarse, or partial information, they can use our method to give larger weight to the less-targeted users and in doing so optimize for market share.
>
> **”Analysis in multi-player settings would make the paper stronger”**
> Further analysis for multiple players is certainly interesting, and it is indeed our hope that this work inspires future work in multi-player settings. We will note though that this is not without challenges. Please see our response to reviewer GVZQ on this matter for more detailed reasoning regarding the challenges involved in extending our results to $n>2$ players.
>
> **”Can the empirical results be provided for realistic applications?”**
> Our empirical results focused primarily on the effectiveness of our method, establishing its robustness and efficiency. That said, we believe our framework lays a strong foundation for future empirical studies, and if applicable, we can address more potential directions for empirical validation in the final version.
>
> **”Absence of error bars in the numerical results”**
> Please note that we report standard errors in the Appendix (Table 5). These were omitted from the main paper for clarity, but if you feel these would be helpful there, we can certainly incorporate them back in.
>
>
>
> Once again, thank you for the detailed feedback. If you found the above clarifications and improvements to your satisfaction, we would greatly appreciate it if you would consider raising your score.
>
>
> [1] Milchtaich, I. (1996). Congestion games with player-specific payoff functions.

---

> > ### Comment · Reviewer_37Jm · 2025-04-04
> >
> > Thank you for the rebuttal and additional experiment on the imperfect information setting. I like the conceptual contributions of the paper and maintain my recommendation for acceptance. It would have been much stronger if the market setting were more realistic ( multiple players, imperfect information, players with different data points, etc.). However, this would require more comprehensive treatment that could be deferred to future work.

---

> > > ### Author Response · Authors · 2025-04-07
> > >
> > > Thank you for your positive feedback and for acknowledging the conceptual contributions. We greatly appreciate your comments and agree that exploring more realistic market settings is an important direction for future work.

---

### Decision · Program_Chairs · 2025-05-01

**Decision:**

Accept (poster)

**Comment:**

The reviewer consensus is that this paper makes an interesting contribution by studying the interplay of competition and model accuracy. They find the paper to be well written and supported.